# Structure and mechanism of a phage-encoded SAM lyase revises catalytic function of enzyme family

Xiaohu Guo[1†‡], Annika Söderholm[1†], Sandesh Kanchugal P[1†], Geir V Isaksen[1,2†], Omar Warsi[3], Ulrich Eckhard[1§], Silvia Trigüis[1], Adolf Gogoll[4], Jon Jerlström-Hultqvist[1,3], Johan Åqvist[1], Dan I Andersson[3], Maria Selmer[1*]

[1]Department of Cell and Molecular Biology, Uppsala University, Uppsala, Sweden; [2]Hylleraas Centre for Quantum Molecular Sciences, Department of Chemistry, UiT - The Arctic University of Norway, Tromsø, Norway; [3]Department of Medical Biochemistry and Microbiology, Uppsala University, Uppsala, Sweden; [4]Department of Chemistry-BMC, Uppsala University, Uppsala, Sweden

**\*For correspondence:**
maria.selmer@icm.uu.se

[†]These authors contributed equally to this work

**Present address:** [‡]Division of Biochemistry, Electron microscopy facility, Cancer Genomics Center, Netherlands Cancer Institute, Amsterdam, Netherlands; [§]Proteolysis Lab, Department of Structural Biology, Molecular Biology Institute of Barcelona, CSIC, Barcelona Science Park, Catalonia, Spain

**Competing interests:** The authors declare that no competing interests exist.

**Abstract** The first S-adenosyl methionine (SAM) degrading enzyme (SAMase) was discovered in bacteriophage T3, as a counter-defense against the bacterial restriction-modification system, and annotated as a SAM hydrolase forming 5'-methyl-thioadenosine (MTA) and L-homoserine. From environmental phages, we recently discovered three SAMases with barely detectable sequence similarity to T3 SAMase and without homology to proteins of known structure. Here, we present the very first phage SAMase structures, in complex with a substrate analogue and the product MTA. The structure shows a trimer of alpha–beta sandwiches similar to the GlnB-like superfamily, with active sites formed at the trimer interfaces. Quantum-mechanical calculations, thin-layer chromatography, and nuclear magnetic resonance spectroscopy demonstrate that this family of enzymes are not hydrolases but lyases forming MTA and L-homoserine lactone in a unimolecular reaction mechanism. Sequence analysis and in vitro and in vivo mutagenesis support that T3 SAMase belongs to the same structural family and utilizes the same reaction mechanism.

## Introduction

S-adenosyl methionine (SAM) plays many important roles in biology. It is an essential methyl donor for methyltransferases that act on nucleic acids, proteins, lipids, and small molecules, but is also involved in many other reactions, for example, as a substrate used in biosynthesis of polyamines and quorum sensing molecules (reviewed in *Loenen, 2006*). It is also involved in epigenetic changes in many organisms (*Janke et al., 2015*; *Su et al., 2016*) and in bacterial restriction-modification systems where methylation of DNA is used to distinguish foreign DNA from host DNA (*Wilson and Murray, 1991*).

In early studies of nucleic acid methylation, it was observed that while infection of *Escherichia coli* with phage T1, T2, and T4 induced higher levels of DNA methylation, infection with phage T3 reduced the degree of methylation of not only DNA, but also tRNA and rRNA (*Gold and Hurwitz, 1964*). The reduction of methylation by T3 infection was associated with an immediate and dramatic lowering of the level of SAM in the *E. coli* cell extract. The degradation products were, using paper chromatography and chemical tests, identified as 5'-methyl-thioadenosine (MTA) and L-homoserine (*Gold and Hurwitz, 1964*; *Figure 1*, top), which led to the claimed discovery of a potent T3-encoded SAM hydrolase enzyme (*Hausmann, 1967*).

Subsequent work showed that the T3 SAMase was produced early in infection (*Gefter et al., 1966*), encoded in the early transcribed portion of the phage genome (*Herrlich and Schweiger,*

**eLife digest** Bacteria can be infected by viruses known as bacteriophages. These viruses inject their genetic material into bacterial cells and use the bacteria's own machinery to build the proteins they need to survive and infect other cells. To protect themselves, bacteria produce a molecule called S-adenosyl methionine, or SAM for short, which deposits marks on the bacteria's DNA. These marks help the bacteria distinguish their own genetic material from the genetic material of foreign invaders: any DNA not bearing the mark from SAM will be immediately broken down by the bacterial cell. This system helps to block many types of bacteriophage infections, but not all. Some bacteriophages carry genes that code for enzymes called SAMases, which can break down SAM, switching off the bacteria's defenses.

The most well-known SAMase was first discovered in the 1960s in a bacteriophage called T3. Chemical studies of this SAMase suggested that it works as a 'hydrolase', meaning that it uses water to break SAM apart. New SAMases have since been discovered in bacteriophages from environmental water samples, which, despite being able to degrade SAM, are genetically dissimilar to one another and the SAMase in T3. This brings into question whether these enzymes all use the same mechanism to break SAM down.

To gain a better understanding of how these SAMases work, Guo, Söderholm, Kanchugal, Isaksen et al. solved the crystal structure of one of the newly discovered enzymes called Svi3-3. This revealed three copies of the Svi3-3 enzyme join together to form a unit that SAM binds to at the border between two of the enzymes. Computer simulations of this structure suggested that Svi3-3 holds SAM in a position where it cannot interact with water, and that once in the grip of the SAMase, SAM instead reacts with itself and splits into two.

Experiments confirmed these predictions for Svi3-3 and the other tested SAMases. Furthermore, the SAMase from bacteriophage T3 was also found to degrade SAM using the same mechanism. This shows that this group of SAMases are not hydrolases as originally thought, but in fact 'lyases': enzymes that break molecules apart without using water.

These findings form a starting point for further investigations into how SAM lyases help bacteriophages evade detection. SAM has various different functions in other living organisms, and these lyases could be used to modulate the levels of SAM in future studies investigating its role.

---

*1970*) and important for the counter defense of the bacteriophage against the type I restriction modification (RM) system of the host bacterium. In this type of RM system, SAM is essential for both

**Figure 1.** SAMase reaction. Top: Hypothetical S-adenosyl methionine (SAM) hydrolase reaction previously suggested to be catalyzed by T3 SAMase. Bottom, green: SAM lyase reaction shown in this study to be catalyzed by all tested bacteriophage SAMases.

The online version of this article includes the following figure supplement(s) for figure 1:

**Figure supplement 1.** Proposed mechanism of rescue of an *ilvA* auxotrophic mutant by SAMases (*Jerlström Hultqvist et al., 2018*).

methylation of host DNA and restriction of target sequences in the foreign DNA. Two potential mechanisms for how the RM system could be impaired were presented. First, the lowered SAM levels prevented methylation of the host genome and the SAM-dependent restriction of the phage genome (*Krueger et al., 1975*). Second, inhibition was observed to be independent of SAM degradation and possibly linked to an interaction with the restriction enzyme (*Spoerel et al., 1979*; *Studier and Movva, 1976*).

Recently, three additional SAM degrading enzymes were identified in a screen for bacteriophage DNA that could rescue an auxotrophic *E. coli* mutant where the isoleucine biosynthetic *ilvA* gene was deleted. Investigations using proteomics and RNA-seq showed that the phage-encoded polypeptides induced up-regulation of the biosynthetic pathway for methionine by degradation of SAM (*Jerlström Hultqvist et al., 2018*), which together with the repressor protein MetJ acts as a co-repressor of the *met* regulon (*Weissbach and Brot, 1991*). Isoleucine biosynthesis was rescued through a promiscuous activity of one of the up-regulated enzymes, MetB (*Figure 1—figure supplement 1*). One of the newly identified SAM degrading enzymes, Svi3-3, was cloned, expressed, and purified. In vitro activity assays demonstrated that Svi3-3 catalyzed conversion of SAM to MTA.

We herein present the first structure of a phage-encoded SAMase and explore the reaction mechanism using both computational and experimental biochemistry. Strikingly, the results unambiguously show that the phage-encoded SAMases are not hydrolases, as believed since the 1960s, but lyases (*Figure 1*).

## Results

### Structure determination of a SAM hydrolase enzyme

The Svi3-3 enzyme was originally expressed from a library of fragmented environmental phage DNA. For this reason, the exact size of the original open reading frame was unknown, but an N-terminally hexahistidine-tagged 162 amino acid construct including some vector-derived sequence was shown to have SAMase activity (*Jerlström Hultqvist et al., 2018*). The phage-derived sequence was subcloned to allow proteolytic removal of the hexahistidine tag, and based on predictions of a disordered N terminus, truncated variants were made. Full-length and N-terminally truncated variants of Svi3-3 were expressed with an N-terminal hexahistidine tag and purified on a Ni-column for structural studies. When expressed from the T7-promoter expression plasmid, the proteins were highly toxic and the tightly regulated, arabinose-inducible BL21-AI cells had to be used to allow cell growth until induction. An N-terminally truncated 146 amino acid construct of Svi3-3, where the hexahistidine tag had been removed (Svi3-3_d19, *Supplementary file 1*-table 1), formed crystals in the presence of SAM or the analogue S-adenosyl homocysteine (SAH). We here number its sequence starting at the N terminus of the TEV-cleaved protein, corresponding to an offset of −16 in relation to previous work (*Jerlström Hultqvist et al., 2018*). The crystals diffracted to 1.45 Å resolution in space group $F4_132$ and the structure was solved using ab initio methods with Arcimboldo (*Rodríguez et al., 2009*; *Figure 2A,F*, *Table 1*). There is one monomer in the asymmetric unit, forming a trimer around the threefold crystallographic axis. Apart from the disordered C-terminal 16 residues, the full structure could be built.

### Structure of Svi3-3

Svi3-3 forms a trimer of ferredoxin-like fold alpha–beta sandwiches, where each subunit has two helices on the outside and a five-stranded anti-parallel beta sheet at the center (*Figure 2A and B*, *Figure 2—video 1*). The subunits pack in a triangular manner, forming continuous beta sheets with their neighbors and are each in a 'velcro-like' topology where residues 2–6 at the N terminus of one subunit form beta-sheet interactions with residues 120–124 in β8 of another subunit (*Figure 2A*). The long β5 forms beta-sheet interactions with both of the other monomers through interactions between the backbones around residues 51 and 55. In addition, the end of the β5–β6 hairpin packs against α1 of another monomer. The trimer interfaces are each stabilized by a single salt bridge, numerous hydrogen bonds, and by hydrophobic interactions at the center.

The substrate analogue SAH binds at the interface between two subunits (*Figure 2C and D*), enclosed by the β5–β6 hairpin at the edge of the sheet. The adenosine base is recognized by interactions with the backbone carbonyl and amide of Ile77 and the side chain of Ser50. The ribose forms

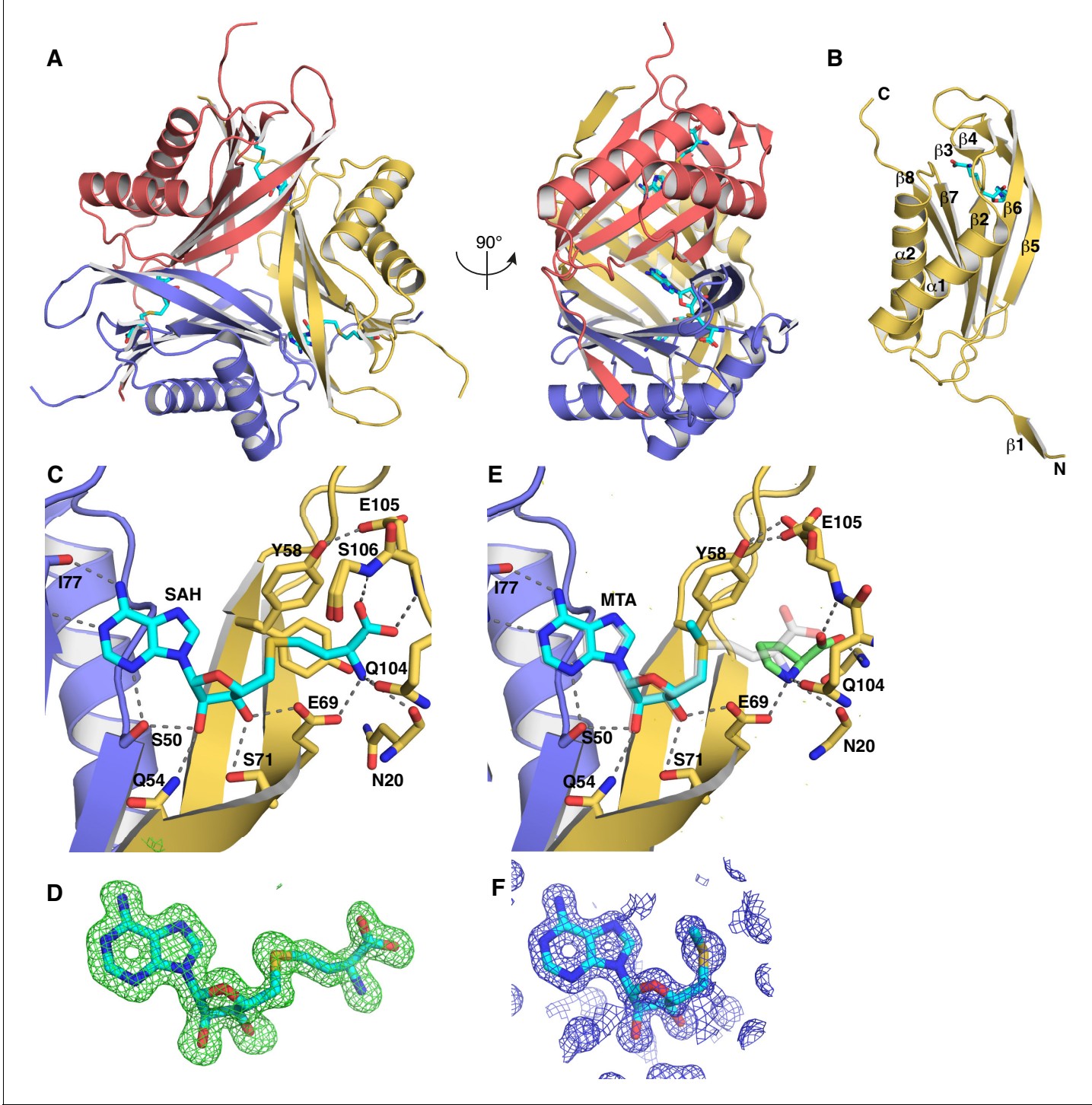

**Figure 2.** Structure of SAMase Svi3-3. (**A**) Structure of the Svi3-3 trimer in complex with S-adenosyl homocysteine (SAH). (**B**) Monomer structure of Svi3-3. (**C**) Interactions of SAH (cyan sticks) at the trimer interface, (**D**) $F_o$–$F_c$ omit map for SAH contoured at three sigma. (**E**) 5'-Methyl-thioadenosine (MTA; cyan sticks) and Pro (green sticks), overlay of SAH is shown in transparent light gray sticks. (**F**) Unbiased Arcimboldo (*Rodríguez et al., 2009*) electron density map for MTA bound to Svi3-3, contoured at two sigma (0.39 e⁻/Å³).

The online version of this article includes the following video for figure 2:

**Figure 2—video 1.** Movie showing the overall structure of the Svi3-3 trimer in complex with S-adenosyl homocysteine (SAH).

https://elifesciences.org/articles/61818#fig2video1

**Table 1.** Crystallographic data and refinement statistics.

| | Svi3-3 MTA | Svi3-3 SAH | Svi3-3 apo |
|---|---|---|---|
| Data collection | | | |
| Beamline | ID23-1 | ID23-1 | ID29 |
| Wavelength | 0.9184 | 0.9184 | 1.0722 |
| Space group | F4$_1$32 | F4$_1$32 | F4$_1$32 |
| Unit cell parameters | | | |
| a, b, c (Å) | 152.3, 154.3, 154.3 | 154.5, 154.5, 154.5 | 158.8, 158.8, 158.8 |
| α, β, χ (°) | 90, 90, 90 | 90, 90, 90 | 90, 90, 90 |
| Resolution (Å)* | 46.54–1.45 (1.54–1.45) | 44.59–1.48 (1.57–1.48) | 39.62–2.8 (2.95–2.80) |
| R$_{meas}$(%)* | 15.2 (86.2) | 12.3 (111.1) | 12.5 (93.9) |
| <I/σ(I)>* | 18.8 (3.9) | 32.44 (2.04) | 9.6 (1.8) |
| CC ½ (%)* | 99.8 (94.8) | 100 (77.4) | 99.3 (61.0) |
| Completeness (%)* | 99.9 (99.7) | 99.7 (98.5) | 97.7 (98.5) |
| Redundancy* | 41.3 (40.2) | 95.27 (24.9) | 5.2 (5.3) |
| Refinement | | | |
| Resolution (Å) | 46.54–1.45 | 44.6–1.48 | 39.36–2.8 |
| Reflections / test set | 28569/1428 | 26716/1338 | 4395/221 |
| R$_{work}$/R$_{free}$ (%) | 13.3/15.5 | 13.9/17.0 | 23.2/27.4 |
| Non-hydrogen atoms | 1282 | 1223 | 992 |
| Protein | 1122 | 1091 | 979 |
| Ligand/ion | 31 | 10 | 10 |
| Water | 129 | 122 | 3 |
| B-factors | 22.8 | 26.0 | 76.6 |
| Protein | 21.8 | 24.9 | 76.7 |
| Ligands | 14.7 | 17.4 | 7.3 |
| Solvent | 33.4 | 36.1 | 57.8 |
| RMSD from ideal | | | |
| Bond lengths (Å) | 0.008 | 0.021 | 0.003 |
| Bond angles (°) | 1.1 | 1.8 | 0.47 |
| Ramachandran plot | | | |
| Preferred (%) | 98.4 | 98.4 | 95.9 |
| Allowed (%) | 0.8 | 0.8 | 4.1 |
| Outliers (%) | 0.8 | 0.8 | 0 |

* Values within parenthesis refer to the highest resolution shell.

interactions with Ser50 from one subunit and Glu69 and Ser71 from the other subunit. The homoserine moiety forms interactions with backbone groups and the side chains of Glu69 and Gln104. In the structure from co-crystallization with SAM, catalysis has occurred and the product MTA is observed in identical position as the corresponding part of SAH, with the methyl group pointing away from Glu69 (*Figure 2E and F*), while a proline molecule from the cryo buffer is bound in the second half of the active site, mimicking the second product. The protein structures are virtually identical in the presence of SAH and MTA (root mean square deviation [RMSD] of 0.16 Å over 129 C$_\alpha$ atoms). To test if the structure contains all the elements needed for activity in vivo, and because the start and end of the native phage protein sequence remain unknown, constructs truncated to only contain the ordered parts were tested for their ability to rescue the *ilvA* knockout mutant. Indeed, the N-terminally truncated construct additionally lacking the disordered C terminus provided similar rescue to full-length Svi3-3, indicating that neither tail is needed for SAMase activity (data not shown). Based

on this result, all further experiments were performed with the 146 amino acid Svi3-3_d19 construct (*Supplementary file 1*-table 1).

Crystallization in the absence of ligands led to a lower-resolution apo structure. The structure of Svi3-3 in apo state is overall very similar to the complex structures (RMSD of 0.84 Å over 124 C$_\alpha$ atoms), but there is a shift of the N terminus, a conformational change of residues 7–9 to form a more extensive interaction with β9 and the preceding loop of the neighboring subunit, leading to a slight loosening of the trimer (*Figure 3*). In addition, residues 64 and 65 in the β5–β6 hairpin are disordered.

Small angle X-ray scattering (SAXS) data confirmed that Svi3-3 has a similar trimeric structure in apo state as in the presence of SAM (*Figure 3—figure supplement 1*), indicating that product release and substrate binding do not require trimer dissociation.

## Similar structures

A search for structures with similar fold and connectivity as Svi3-3 using PDBeFold (*Krissinel and Henrick, 2004*) showed that bacterial PII signaling proteins and cation tolerance proteins of CutA type (belonging to a PII-like family) in the GlnB-like superfamily have similar trimeric architectures of ferredoxin-like folds. The closest structural neighbors superpose with RMSDs above 2 Å over no more than 86 C$_\alpha$ atoms. PII regulatory proteins bind ATP/ADP and 2-oxoglutarate at the trimer interface, and conformational changes of loops in response to ligand binding and modification modulates binding to other proteins (*Forchhammer and Lüddecke, 2016*). Intriguingly, the position of ATP is similar to where SAH binds to Svi3-3, although the binding site appears non-conserved.

## Catalytic residues

The reaction mechanism of the presumed SAM hydrolases (*Figure 1*) is previously unexplored. Two possibilities would either be that the enzyme provides a catalytic base that by deprotonation activates a water molecule for nucleophilic attack on the γ carbon of SAM, or that a catalytic residue from the enzyme acts as nucleophile and attacks the γ carbon, forming a covalent intermediate with the substrate that subsequently gets hydrolyzed by reaction with a water molecule. In both scenarios, a catalytic amino acid in proximity to the site of hydrolysis would be needed.

Inspection of the active site shows that Glu69 is relatively close to the site of cleavage (*Figure 2C*), making this residue the most likely candidate. Another possible residue that could act

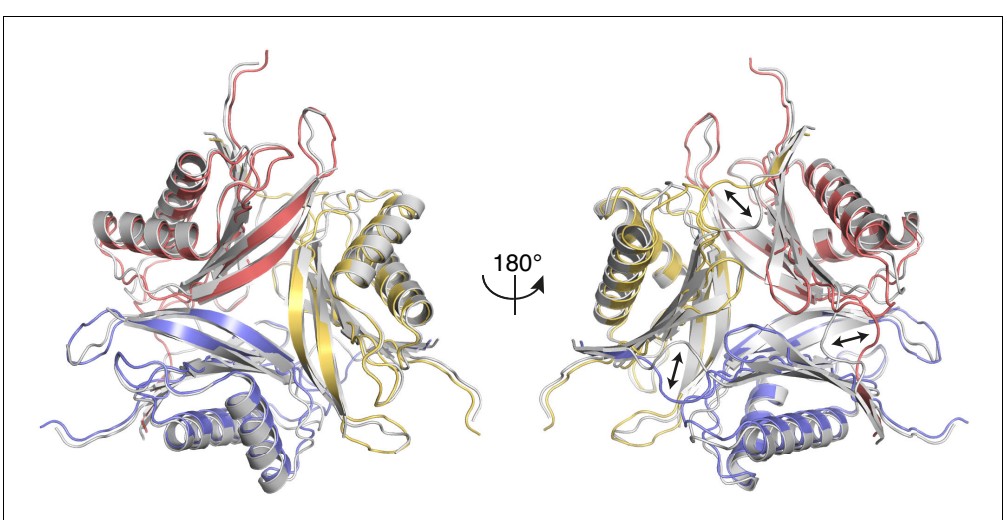

**Figure 3.** Comparison of the apo structure of Svi3-3 (gray) and the 5'-methyl-thioadenosine (MTA) complex structure (colored as in Figure 2A, transparent). Arrows indicate conformational differences between the N-terminal regions in the two structures.

The online version of this article includes the following figure supplement(s) for figure 3:

**Figure supplement 1.** Size-exclusion chromatography coupled to small angle X-ray scattering (SEC-SAXS) of Svi3-3.

as a base or a nucleophile is Tyr58, which in such case would need to donate a proton to the nearby Glu105 (*Figure 2C*). To test these possibilities, Svi3-3 mutants Y58F, E69Q, E69A, and E105Q were constructed, expressed, and purified for in vitro activity assays. The Y58F mutant eluted mainly as monomer, indicating a de-stabilized trimer, while all other mutants migrated as trimers. The activity of wild-type (WT) and mutant Svi3-3 variants in conversion of SAM to MTA was determined at 1 mM substrate concentration. The WT truncated Svi3-3 construct showed an average turnover of 9.5 s$^{-1}$ at 25°C, nearly doubled compared to the full length, His-tagged construct (*Jerlström Hultqvist et al., 2018*), but with significant deviation between batches. The mutant enzymes all showed reduced activity compared to WT (*Figure 4A*, *Figure 4—figure supplement 1*). For the Y58F and E105Q mutants, activity was fivefold and threefold reduced, whereas for the E69Q and E69A mutants, activity was nearly abolished (4500- and 1500-fold reduction). Thus, Glu69 is in vitro critical for substrate binding, catalysis, or both.

## Thermal shift binding assays

To characterize the binding of substrate and product to Svi3-3, thermal unfolding experiments were performed using differential scanning fluorimetry (DSF) (*Niesen et al., 2007*). In the presence of the substrate SAM, its analogue SAH, or the reaction product MTA, the melting temperature of the WT enzyme increased by up to 30°C, demonstrating a major stabilizing effect on the structure (*Figure 4B*). Under the assay conditions, the substrate is most likely turned over, and as a result, the sample with SAM is stabilized by binding of MTA. A large stabilization was also seen in thermal denaturation assays followed by circular dichroism in the presence of MTA (data not shown).

The DSF assay clearly demonstrates that the E69Q mutant is not impaired in binding of SAM or MTA, but that binding of SAH is abolished (*Figure 4B*). However, despite extensive trials, a structure of Svi3-3 E69Q with SAM could not be obtained. A crystal structure of Svi3-3 E69Q with MTA shows no major conformational changes of the enzyme, but a minor shift of the mutated side chain close to the reaction site (data not shown).

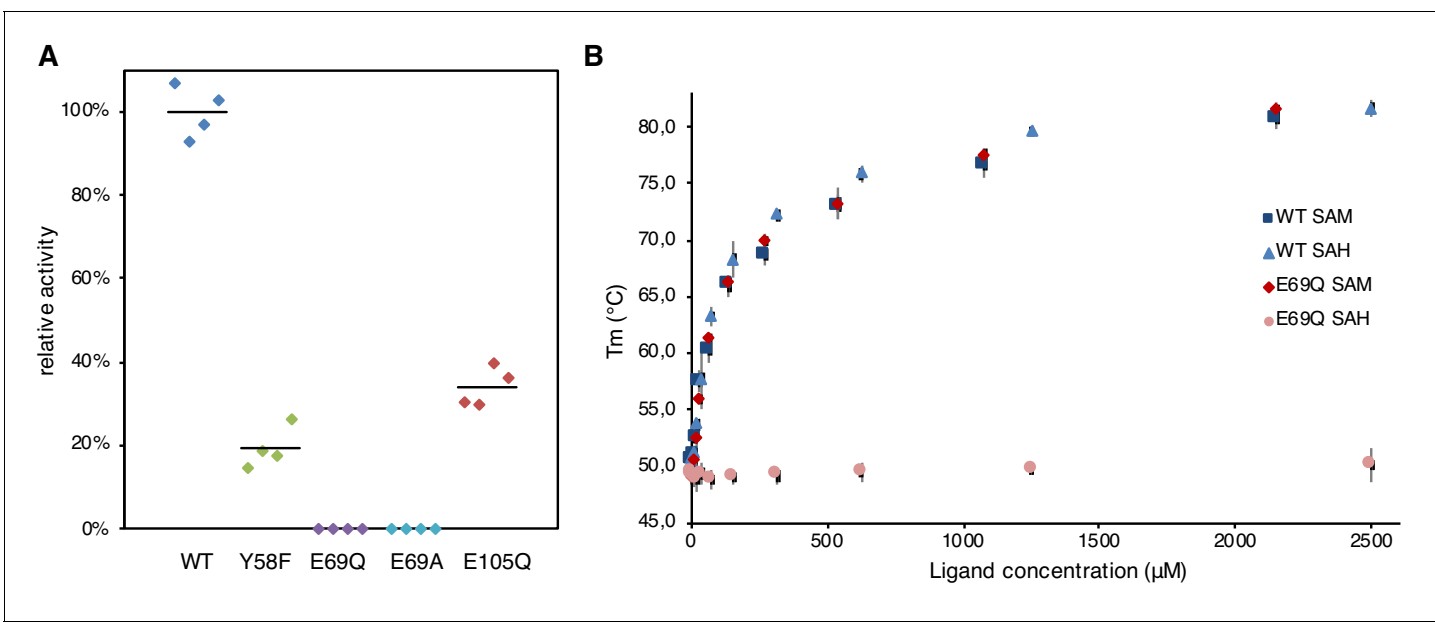

**Figure 4.** Svi3-3 assays. (**A**) Relative enzymatic activity of Svi3-3 variants. The data points are based on technical duplicates from two different protein purifications, where activity is related to wild type (WT) purified at the same time and the average is represented by a black line. (**B**) Differential scanning fluorimetry (DSF) data for Svi3-3 variants with different concentrations of S-adenosyl methionine (SAM) and S-adenosyl homocysteine (SAH). Error bars correspond to ±1 standard deviation based on triplicate data.

The online version of this article includes the following figure supplement(s) for figure 4:

**Figure supplement 1.** Absolute enzymatic activity of Svi3-3 variants.

## Structure-guided sequence alignment supports conservation of structure and reaction mechanism

A structure-guided multiple sequence alignment of Svi3-3 with T3 SAMase and two other polypeptides annotated with SAMase activity (Orf1 and Svi3-7 [*Jerlström Hultqvist et al., 2018*]) but with barely detectable sequence similarity suggests that a similar core fold can be formed by all aligned enzymes (*Figure 5*). Only three amino acids are strictly conserved, and only a handful conservatively substituted. Based on the SAH structure, all the conserved residues are important for binding of SAM. Gly56 packs against the ribose, not allowing space for a side chain, Glu69 and Gln104 as described above form hydrogen bonds to the ligand, and Tyr58 packs against the γ carbon and forms a hydrogen bond to Glu105, enclosing the ligand.

The Svi3-3 structure allows interpretation of the effect of loss-of-function mutations identified in the previous complementation studies (*Jerlström Hultqvist et al., 2018*). One group of disabling mutations would act to sterically disturb the interaction between the monomers and/or change the character of the interface (R14C, V33D, V52D, E110K, and G115V). Mutation of the conserved glycine (G56D) would clash with the ribose in MTA and SAH and thereby disturb substrate binding. The remaining mutations (A13V, A93G, and G95D) are likely to affect folding or stability of the structure.

## Mutagenesis of catalytic residues in homologous enzymes

To validate the structure-guided sequence alignment, Glu68 in T3 SAMase, predicted by the sequence alignment (*Figure 5*) to be the equivalent of Glu69 in Svi3-3, was mutagenized and tested for in vitro activity. The E68Q mutant of T3 SAMase retained 20–30% activity compared to WT.

In addition, mutant variants carried on an inducible plasmid were tested for their ability to rescue the ΔilvA auxotrophic mutant (*Figure 1—figure supplement 1*; *Jerlström Hultqvist et al., 2018*) under uninduced and induced conditions. Because of its toxicity at higher expression level, WT T3 SAMase only shows rescue under uninduced conditions (*Jerlström Hultqvist et al., 2018*). Both T3 SAMase and Orf1 contain two consecutive acidic residues (E67 and E68 in T3 SAMase, E50 and D51 in Orf1, *Figure 5*), both of which were conservatively mutated. No rescue was observed either for the Svi3-3 E69Q, Svi3-3 E69A, Orf1 E50Q, or the T3 SAMase E68Q mutants, suggesting a lost or reduced activity of these variants (*Table 2*) and validating the sequence alignment. Mutations of the

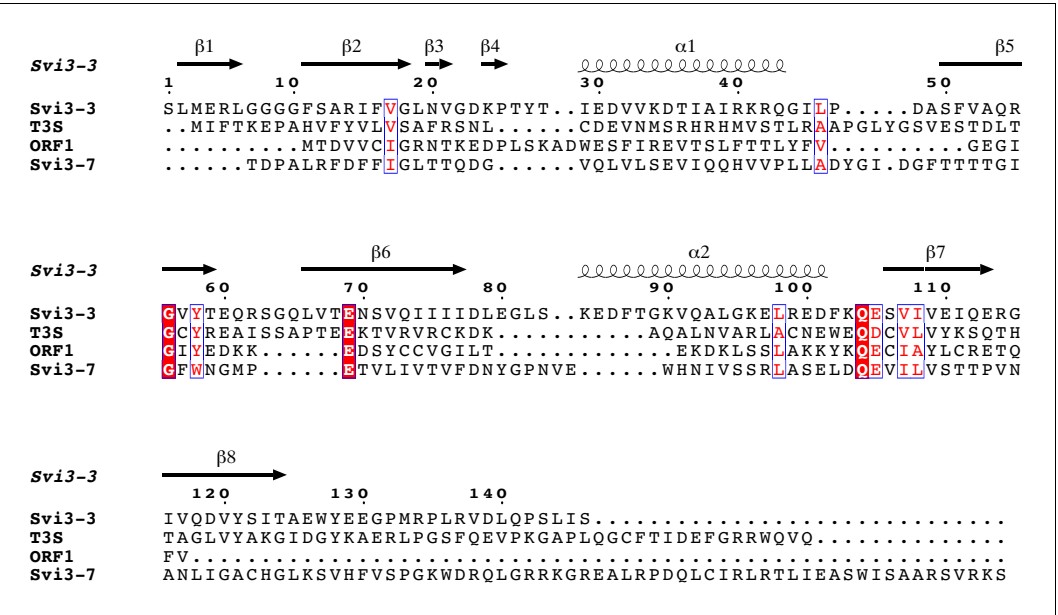

**Figure 5.** Structure-guided sequence alignment of SAMases with demonstrated activity (*Jerlström Hultqvist et al., 2018*). Secondary structure of Svi3-3 is displayed above the alignment. Red boxes with white letters indicate conserved residues and red letters in white boxes show conservatively substituted residues. Figure was prepared using ESPript (*Gouet et al., 2003*).

**Table 2.** In vivo complementation assay using different variants of Svi3-3, Orf1, and T3 SAMase. The SAMase-encoding genes were inserted into the plasmids pCA24N –gfp (Svi3-3 and Orf1) or pRD2 (T3 SAMase) and expressed under control of an isopropyl β-D-1-thiogalactopyranoside (IPTG) inducible promoter.

| Strain | Genotype | SAMase variant | Growth in M9 minimal media | |
| | | | No IPTG | 1 mM IPTG |
| --- | --- | --- | --- | --- |
| DA5438 | WT | - | + | + |
| DA58128 | ΔilvA pRD2 | - | - | - |
| DA48932 | ΔilvA pCA24N::svi3-3 | Svi3-3 wt | - | + |
| DA57022 | ΔilvA pCA24N::svi3-3 | Svi3-3 E69Q | - | - |
| DA57021 | ΔilvA pCA24N::svi3-3 | Svi3-3 E69A | - | - |
| DA51653 | ΔilvA pCA24N::orf1 | Orf1 wt | - | + |
| DA67997 | ΔilvA pCA24N::orf1 | Orf1 E50Q | - | - |
| DA67998 | ΔilvA pCA24N::orf1 | Orf1 D51N | - | + |
| DA57899 | ΔilvA pRD2::t3 | T3 SAMase wt | + | - |
| DA58126 | ΔilvA pRD2::t3 | T3 SAMase E67Q | + | - |
| DA58127 | ΔilvA pRD2::t3 | T3 SAMase E68Q | - | - |

neighboring acidic residues (E67Q in T3 SAMase and D51N in Orf1) still allowed rescue under the same conditions as the corresponding WT.

Given that we observed measurable activity for the T3 SAMase E68Q mutant in vitro, we decided to insert different variants of the T3 SAMase on the chromosome, to allow titration of the expression levels. To this end, the different mutant gene variants were placed downstream of the p*BAD* promoter in a ΔilvA knockout mutant. Different concentrations of L-arabinose were used to induce the expression of these variants to determine if the variants could show in vivo activity at higher expression levels. For the E68Q mutant, growth was only observed at the highest concentration of arabinose (0.1%), where low enzymatic activity can be compensated by increased enzyme levels (*Supplementary file 1*-table 2).

## Molecular dynamics simulations of Svi3-3 in complex with substrate

To gain further insights into substrate binding to Svi3-3, molecular dynamics (MD) simulations were performed. An initial complex of Svi3-3 with SAM was generated from docking calculations, which indicated a slight binding preference for SAM over SAH. A total of 100 ns of MD simulation was run for both apo Svi3-3 and the SAM complex (*Figure 6*). The simulation of the complex revealed that the SAM conformation in the active site is very stable, particularly the methionine part (*Figure 6—figure supplement 1*). The time evolution of the backbone RMSD with respect to the initial structure is shown in *Figure 6A* and reaches stability in about 30 ns. In agreement with the experimental structure with SAH, the carboxylate group is strongly stabilized by the backbone amide hydrogens of Glu105 and Ser106, whereas the amino group is primarily stabilized by the side-chains of Glu69 and Gln104 (*Figure 6—figure supplement 2A*). Moreover, the positively charged methionine S atom is consistently stabilized through a π-cation interaction with the Tyr58 side-chain (*Figure 6—figure supplement 1*, *Figure 6—figure supplement 2A*). The adenosine part of SAM is primarily stabilized by H-bond interactions with the side-chain of Ser50 and the backbone of Val57 and Ile77 (*Figure 6—figure supplement 1*), as in the crystal structures with SAH and MTA (*Figure 2C and E*). Comparison of the average root mean square fluctuation (RMSF) of the protein backbone shows that Svi3-3 is more flexible throughout the 100 ns MD simulation in apo state (RMSF for trimer 0.91 Å$^2$) than with SAM in the active site (RMSF for trimer 0.86 Å$^2$) (*Figure 6B*). Interestingly, the residues surrounding the active site cavity, and in particular the methionine-stabilizing residues 104–106, become significantly more flexible in the apo simulations (*Figure 6B*). Thus, 100 ns of MD simulation demonstrated that apo Svi3-3 shows the highest flexibility in the regions surrounding the active site, and that these regions become significantly more rigid upon substrate binding.

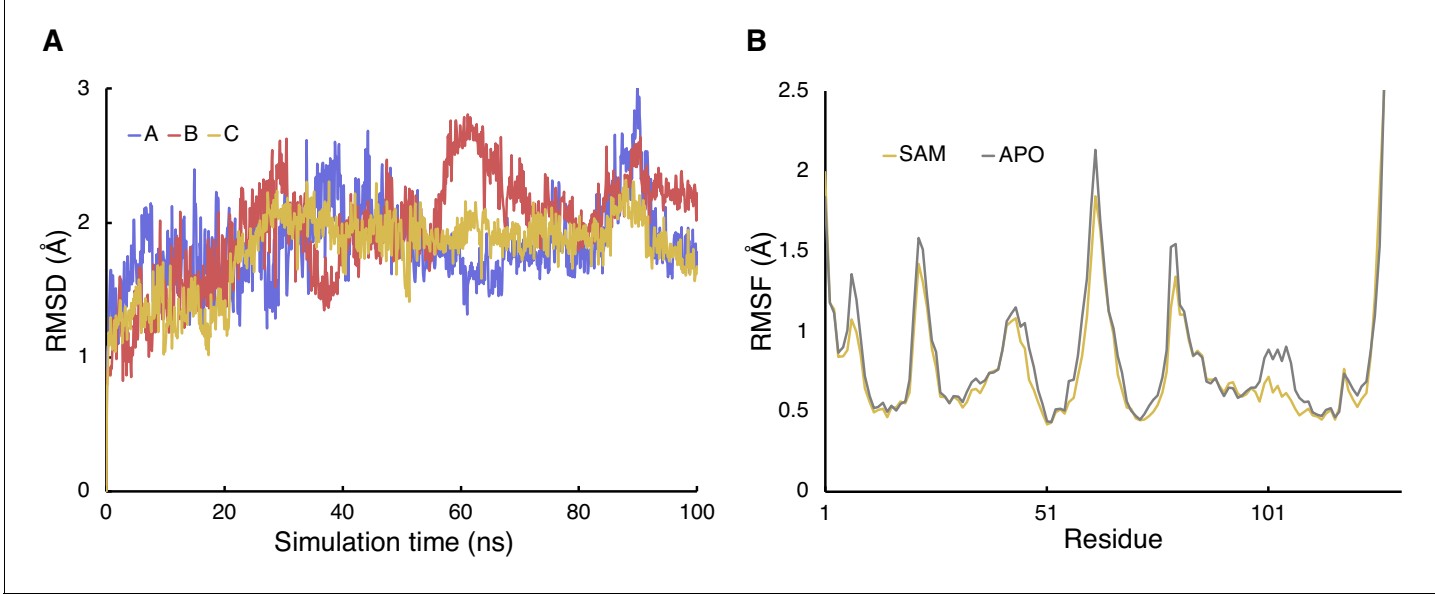

**Figure 6.** Molecular dynamics (MD) simulations of Svi3-3. (**A**) Time evolution of root mean square deviation (RMSD) to the Svi3-3 crystal structure for each monomer during the 100 ns MD simulation in the presence of S-adenosyl methionine (SAM). (**B**) Average backbone root mean square fluctuation (RMSF) over 100 ns MD simulation of apo Svi3-3 and its complex with SAM. The largest difference is seen for active-site residues 104–106. The online version of this article includes the following figure supplement(s) for figure 6:

**Figure supplement 1.** Protein–ligand interactions obtained from 100 ns molecular dynamics (MD) simulations illustrating the percentage of interaction from the total simulation time.

**Figure supplement 2.** Active-site structure of Svi3-3 as observed from MD simulations.

## Quantum mechanical studies of the SAMase reaction mechanism

The mechanism of enzymatic SAMase activity is previously unexplored. Thus, the observed active site conformations from the MD simulations were used to build a 194-atom cluster model (*Figure 6— figure supplement 2B*) to investigate the SAMase reaction mechanism with density functional theory (DFT). The simulations revealed one potential, but not optimally positioned, water molecule for hydrolysis that was stabilized by H-bonds between the backbone of Ser106 and Gln104, but there were no obvious residues within reach to activate this water by acting as a base.

Glu69 was initially suspected to either work as a catalytic base, activating a water molecule for hydrolysis, or as a nucleophile, attacking the γ-carbon of SAM. However, no water molecule was observed with a suitable position with respect to Glu69 in the experimental structures or after the MD simulation, indicating that the role as a catalytic base is unlikely. Moreover, the observed SAM configuration relative to Glu69 was not optimal for attack on the γ-carbon of SAM. DFT geometry optimizations furthermore failed to locate any stationary point (transition state) for the attack, suggesting that the role as a nucleophile is also unlikely. Thus, it seems that the main role of Glu69 is to bind and orient the substrate in the active site by accepting H-bonds primarily from the amino group of SAM, but also from the 3'-hydroxyl group of the ribose ring (*Figure 6—figure supplement 2*). In fact, the only residue in the active site oriented properly for a potential hydrolase reaction mechanism is Tyr58. However, DFT calculations could exclude the possibility of Tyr58 acting as a nucleophile in such a mechanism.

Further DFT geometry optimization, however, revealed a completely different mechanism, a unimolecular reaction resulting in the formation of homoserine lactone (*Figure 7A*). The DFT optimized stationary points shown in *Figure 7A* indicate that Tyr58 is deprotonated and bridged by a water molecule to the protonated Glu105. Deprotonation of Tyr58 enhances its cation-π interaction with the S atom of SAM and the electrostatic preorganization weakens the bond to the ribose ring. Together with the amino group interaction with Glu69 this makes the configuration of SAM in the active site of Svi3-3 susceptible to intramolecular carboxylate oxygen attack on the γ-carbon. The

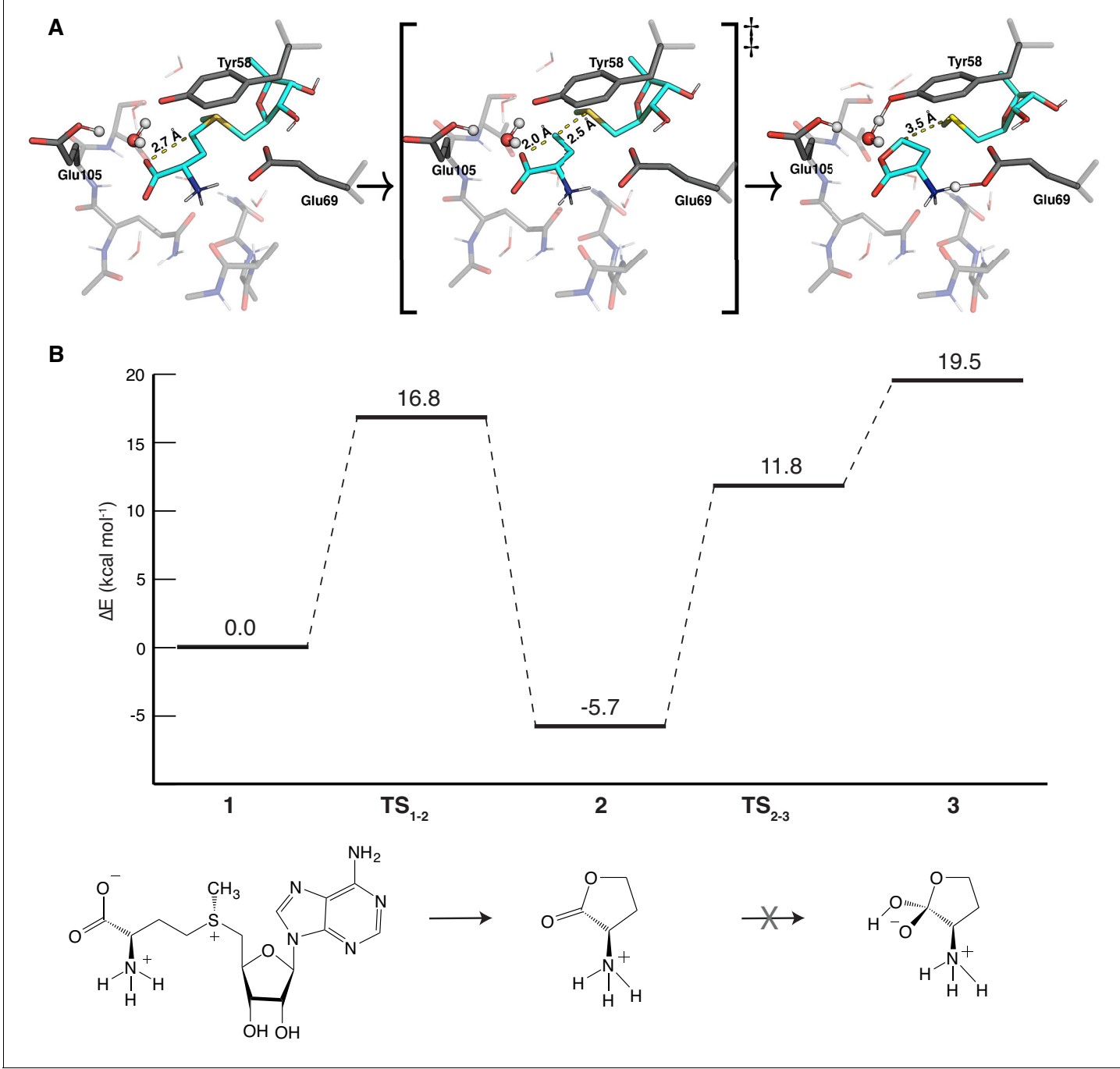

**Figure 7.** Density functional theory (DFT) calculations on the S-adenosyl methionine (SAM) lyase reaction of Svi3-3. (**A**) Optimized DFT structures for the reactant (left), transition (middle), and product (right) state in the SAM lyase reaction mechanism. (**B**) Calculated free energy profiles for reactant (1), transition state (1–2) and product state (2) for the SAM lyase reaction mechanism. Further conversion from homoserine lactone (2) to the tetrahedral intermediate (3) that would form homoserine is not supported by the high DFT energies.

activation energy for the formation of homoserine lactone in the DFT cluster model was calculated to be 16.8 kcal/mol with an exothermic reaction energy of −5.7 kcal/mol (*Figure 7B*). These values were rather insensitive to the choice of dielectric constant, as a change from ε = 4 to ε = 80 yields an energy barrier of 16.1 kcal/mol and a reaction energy of −6.3 kcal/mol. However, since our active site cluster model by necessity is limited in size, it seems possible that the predicted net proton transfer from Tyr58 to Glu105 could be caused by the DFT optimization disfavoring ion pairs with

charge separation over a larger distance (Glu-…TyrOH…S+), and instead moving the proton from Tyr58 to Glu105 (the initial configuration had the proton on Tyr58). In order to examine this issue, the reaction energetics were recalculated with the proton constrained to remain on Tyr58, maintaining the standard protonation states of the tyrosine and Glu105. This, in fact, yielded very similar energetics with a barrier of 15.5 kcal/mol and a reaction energy of −6.1 kcal/mol (ε = 4), indicating that the proton relay is not a necessary feature of the reaction mechanism. In the product state (*Figure 7A*), Glu69 shares a proton with the amino group of SAM, and a water molecule makes strong hydrogen bonds with Tyr58 and Glu105, irrespective of protonation state at the start of the reaction. This water appeared to be a good candidate for nucleophilic attack on the carbonyl carbon of homoserine lactone, thereby forming a tetrahedral intermediate which could break down to homoserine. The calculated barrier for this step is 17.5 kcal/mol, which is not unrealistically high, but the reaction energy for the tetrahedral intermediate is 25.2 kcal/mol relative to the homoserine lactone (*Figure 7B*). Thus, the DFT calculations indicate that Svi3-3 forms homoserine lactone, but that further breakdown to homoserine does not occur in the active site.

Most importantly, the DFT calculations clearly predict that Svi3-3 is not a SAM hydrolase, but rather a SAM lyase, catalyzing the unimolecular transformation of SAM to homoserine lactone, whereafter this product is released from the active site.

## The carboxyl group of SAM appears essential for Svi3-3 binding

One prediction from the suggested reaction mechanism is that the carboxyl group of SAM is essential for the degradation of SAM by Svi3-3. To test this hypothesis, decarboxylated SAM (dcSAM) was produced enzymatically from SAM using SAM decarboxylase (*Cohn et al., 1983*) and used as a substrate in the MTA formation assay. In support of the proposed mechanism, we observed no Svi3-3-catalyzed production of MTA from dcSAM (*Figure 8*). However, in a DSF binding assay, dcSAM does not induce a thermal stabilization of Svi3-3 (*Figure 8—figure supplement 1*). Comparison with

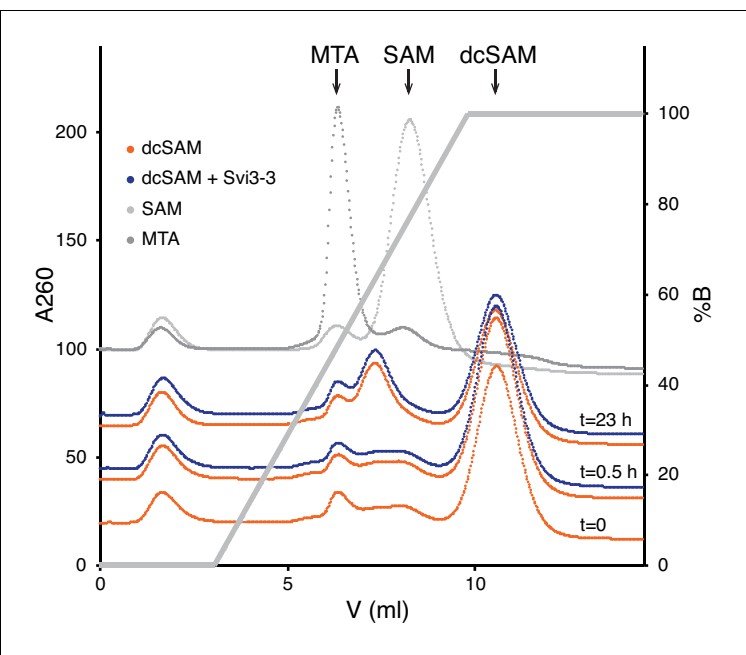

**Figure 8.** Representative chromatogram from ion exchange chromatography of decarboxylated S-adenosyl methionine (dcSAM) reactions and controls. 0.32 mM dcSAM mix (77% dcSAM) was incubated ± Svi3-3 for 0–23 hr.

The online version of this article includes the following figure supplement(s) for figure 8:

**Figure supplement 1.** Differential scanning fluorimetry (DSF) duplicate data for Svi3-3 in the presence of different concentrations of decarboxylated S-adenosyl methionine (dcSAM) mix (72% dcSAM, 28% S-adenosyl methionine [SAM] + 5'-methyl-thioadenosine [MTA], orange plus signs) and SAM (open blue squares).

the strong stabilization by SAM/MTA and SAH (*Figure 4B*) suggests that the carboxyl group of SAM is required for binding to Svi3-3.

## Experimental verification of reaction products by thin-layer chromatography and nuclear magnetic resonance spectroscopy

Our activity assay (*Jerlström Hultqvist et al., 2018*) is based on detection of MTA that is also observed in the structure from co-crystallization of Svi3-3 with SAM. Thus, MTA is indeed formed in the reaction. In order to test the predictions from the DFT calculations, we used thin-layer chromatography (TLC), commonly used for separation of amino acids, to determine whether homoserine was formed. The substrate SAM and the products could be separated by TLC, the adenosyl-containing compounds were visualized under UV light, and the amines were stained with ninhydrin after pre-treatment with β-mercapto ethanol (BME). Interestingly, we observed a reaction product with mobility and color upon staining distinct from homoserine, proving that Svi3-3 is not a SAM hydrolase. The same assay was performed with T3 SAMase and Orf1 and comparison with reference samples shows that for all three enzymes, a reaction product is observed that on TLC migrates and stains similar to homoserine lactone (*Figure 9A*), supporting the hypothesis that Svi3-3, T3 SAMase and Orf1 are indeed SAM lyases.

To confirm the identity of the reaction product, the enzymatic reaction products for Svi3-3 were examined with $^1$H nuclear magnetic resonance (NMR) and the spectrum compared to reference spectra for SAM, homoserine, and homoserine lactone (*Figure 9B*, *Figure 9—figure supplement 1*, *Figure 9—figure supplement 2*). The results confirm that homoserine lactone is formed on the same time scale as MTA. Upon incubation of homoserine lactone in phosphate buffer at pH 7.4, the lactone is transformed to homoserine through spontaneous hydrolysis (*Figure 9—figure supplement 3*).

## Discussion

### Relation to other enzymes with similar activity and structure

Svi3-3 represents the first structure of a phage-encoded SAM degrading enzyme. There is no previously established reaction mechanism for these enzymes, and to our knowledge there are only two previous examples of enzymes that can cleave a trialkyl sulfonium substrate. A distinct type of SAM lyase, 1-aminocyclopropane-1-carboxylate synthase (ACC synthase, EC 4.4.1.14) exists in higher plants and some fungi. The products of this enzyme are MTA and 1-aminocyclopropane-1-carboxylate that is used in the biosynthetic pathway for ethylene. The reaction is PLP-dependent and the enzyme is structurally and mechanistically unrelated to the phage encoded SAM lyases (*Capitani et al., 1999*).

Instead, Svi3-3 shows structural similarity to PII and PII-like proteins, many of which bind nucleotides or nucleotide-derived metabolites in the inter-subunit clefts. Despite very low levels of sequence identity within those families, they have been suggested to have arisen by divergent evolution from a common ancestor (*Forchhammer and Lüddecke, 2016*). Based on this similarity, we tested binding of ATP, ADP, and AMP to Svi3-3 by DSF, but detected no interaction. Still, these two families of enzymes may have a distant evolutionary relationship.

### Substrate binding

Svi3-3 forms a trimer in solution both in the absence and in the presence of substrate (*Figure 3—figure supplement 1*). Thermal shift binding experiments showed a major stabilization of Svi3-3 upon binding of MTA or SAH (*Figure 4B*). Comparisons of the apo and ligand-bound structures (*Figure 3*) as well as MD simulations (*Figure 6B*) indicate that stabilization is caused by tightening of the trimer around the ligand. Results presented here show that E69 in Svi3-3 plays a critical role in the enzyme, and the failure in getting crystals of an E69Q mutant with SAM suggests that binding of SAM may be associated with conformational changes that are incompatible with crystal packing or with the crystallization conditions. The observation that the E69Q mutant binds SAM but does not bind SAH indicates that although SAH only lacks one methyl group compared to SAM, the complex structure of WT Svi3-3 with SAH may not fully mimic the substrate-bound state. Since MTA and SAH show a perfect overlap between the two structures (*Figure 2D*), any such difference in binding mode

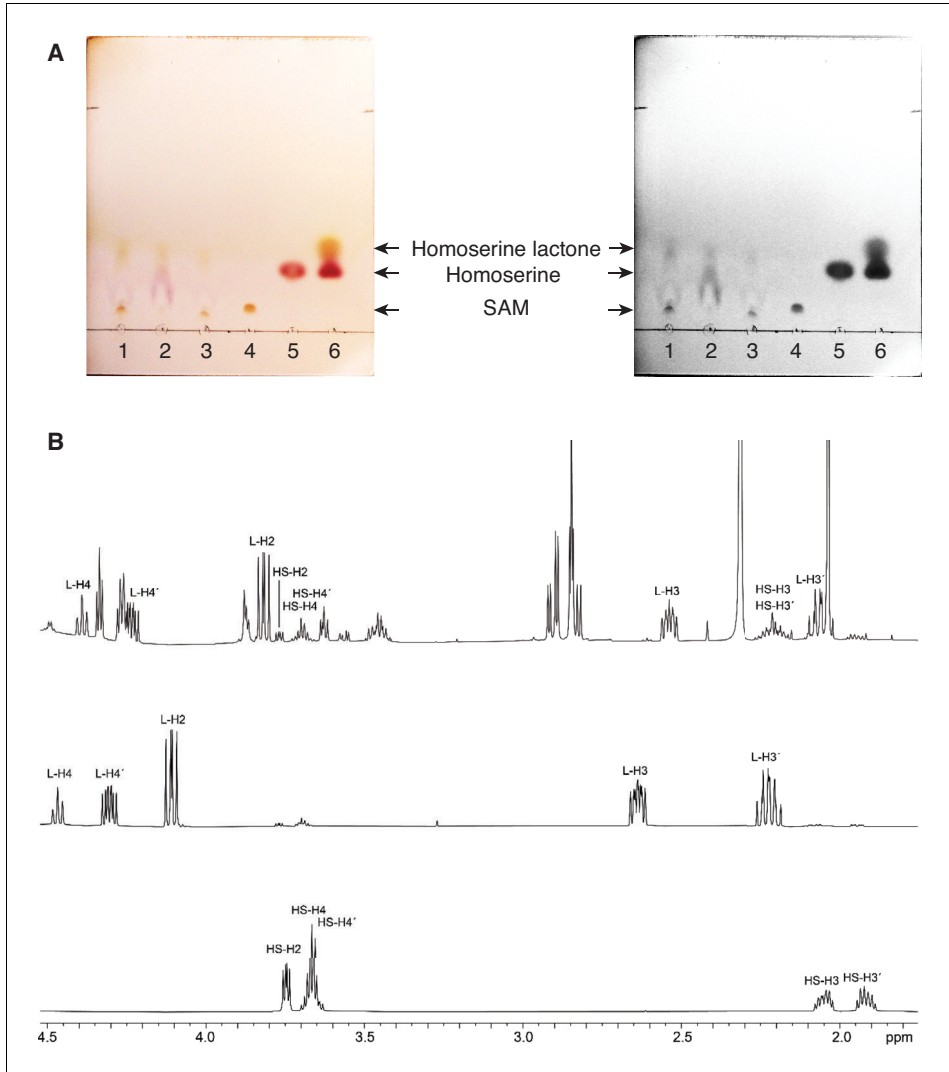

**Figure 9.** Characterization of SAMase reaction products. (**A**) TLC separation of enzymatic reactions and controls, shown in color and gray scale for clarity (1: Svi3-3, 2: T3 SAMase, 3: Orf1, 4: S-adenosyl methionine [SAM; 17 nmol], 5: L-homoserine [20 nmol], 6: L-homoserine lactone [400 nmol]). (**B**) $^1$H nuclear magnetic resonance (NMR) spectra at 600 MHz in sodium phosphate buffer, D$_2$O, pH 7.4. Top: Enzymatic degradation of SAM (4 mM) (500 nM enzyme) after 45 min. Middle: Homoserine lactone (59 mM) showing onset of hydrolysis after 10 min. Bottom: Homoserine (57 mM).

The online version of this article includes the following figure supplement(s) for figure 9:

**Figure supplement 1.** $^1$H nuclear magnetic resonance (NMR) spectra of a reference sample of homoserine lactone (600 MHz, 59 mM in sodium phosphate buffer, D$_2$O, pH 7.4).

**Figure supplement 2.** $^1$H nuclear magnetic resonance (NMR) spectra of a reference sample of homoserine (600 MHz, 57 mM in sodium phosphate buffer, D$_2$O, pH 7.4).

**Figure supplement 3.** $^1$H nuclear magnetic resonance (NMR) spectra of a reference sample of homoserine lactone at various concentrations and exposure times to sodium phosphate buffer showing progressive hydrolysis to homoserine (600 MHz, sodium phosphate buffer, D$_2$O, pH 7.4).

between SAM and SAH is likely to involve the methionine end of the substrate, where MD indicates that Tyr58 forms a cation-π interaction with the positively charged sulfur. The reason why the E69Q mutant does not bind to SAH may be that two important interactions are lacking; Q69 cannot accept two hydrogen bonds (*Figure 2C*) and there is no positive charge on the sulfur that can participate in the cation-π interaction with Tyr58 (*Figure 6—figure supplement 2*).

## Phage-encoded SAMases are lyases

Prompted by DFT calculations producing high energy barriers for a hydrolysis reaction within the active site of Svi3-3 (*Figure 7B*), the TLC assays and NMR show unambiguously that Svi3-3 is a SAM lyase forming homoserine lactone and MTA (*Figure 9*). The lactone is spontaneously converted to homoserine in solution (*Wu et al., 1983*; *Figure 9—figure supplement 3*, *Figure 10*). Previous attempts to set up a coupled SAM hydrolase activity assay for Svi3-3 using homoserine dehydrogenase or homoserine kinase failed to show homoserine production with the same rate as MTA production (data not shown). The observed rates were 1000-fold lower, and we can now explain that this is due to the slow and un-catalyzed formation of homoserine from the homoserine lactone that is enzymatically formed.

Computational work by others suggests that the non-enzymatic degradation of SAM to homoserine lactone and MTA is slowed down by the favorable interactions of the carboxylate group with water (*Lankau et al., 2017*). Thus, Svi3-3 increases the reactivity of the carboxylate group by excluding water from the corresponding part of the active site, while stabilizing a reactive conformation of the substrate. For this unimolecular reaction mechanism (*Figure 10A*), only very few strictly conserved residues are required, as illustrated in the multiple-sequence alignment (*Figure 5*). Both hydrogen bond acceptors of Glu69 seem critical for stabilization of the reactive state (*Figure 10*), and Svi3-3 E69Q has nearly abolished activity. However, the effect of the corresponding mutation in T3 SAMase is not quite as dramatic. Since the level of sequence identity between the two enzymes is low, there will be many differences in the active site, and additional interactions may contribute to stabilizing the reactive state for the same mechanism in T3 SAMase.

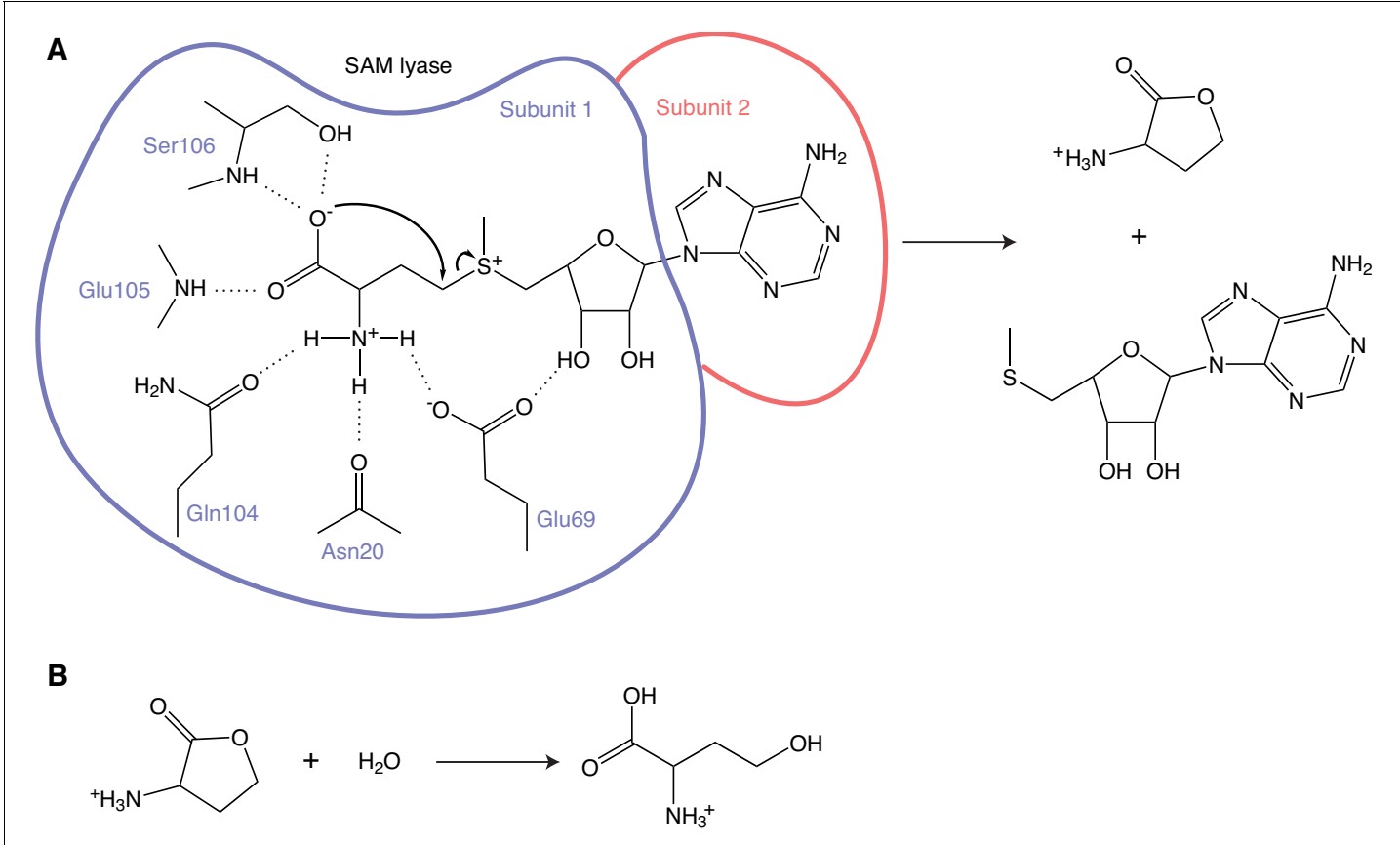

**Figure 10.** Mechanism of S-adenosyl methionine (SAM) degradation. (**A**) The first step is catalyzed by the SAM lyase, releasing the products homoserine lactone and 5'-methyl-thioadenosine (MTA). Hydrogen bonds between the enzyme and the reacting part of SAM are shown as dashed lines. The two subunits contributing to one active site are indicated in slate and salmon. (**B**) Homoserine lactone is spontaneously hydrolyzed to homoserine in solution.

The only previously partly characterized phage SAMase comes from phage T3 (*Hausmann, 1967*). It was identified based on its anti-restriction activity, but the mechanism was never fully clarified. Anti-restriction activity of the closely related T7 phage is based on the OCR protein that forms a structure that mimics B-form DNA and blocks DNA binding of EcoKI and other type I RM systems (*Walkinshaw et al., 2002*). The first structure from the SAM lyase enzyme family that we present here clearly proves that they have no structural similarity to the OCR protein. Instead, our data shows that also the T3 SAMase is a lyase and not a hydrolase, but future studies are needed to elucidate whether these enzymes also have additional mechanisms of anti-restriction activity (*Spoerel et al., 1979*).

Around the same time as the SAM-degrading enzyme from bacteriophage T3 was discovered, the same enzymatic activity was also found in extract from bacteria (*Shapiro and Mather, 1958*) and yeast (*Mudd, 1959a*; *Mudd, 1959b*). In both of these systems, the reaction products were described as MTA and γ-aminobutyro-lactone (homoserine lactone), and the conversion to homoserine was considered to be spontaneous. In contrast, the phage enzyme from T3 was early described as a SAM hydrolase, and referred to as such until this day. In the early literature, homoserine lactone was identified as an intermediate (*Gold et al., 1964*) but, perhaps due to the available methods at the time, it was not realized that homoserine was formed on a different time-scale from MTA, indicating a spontaneous and not enzyme-catalyzed reaction. For this reason, it is not until now that we can correct the functional annotation to SAM lyase.

The SAM lyases show very low sequence conservation and large variations in size (*Figure 5*), and future studies will elucidate the relationship between structure and activity in this family of enzymes, their prevalence, and their exact biological roles in different organisms.

## Materials and methods

### Cloning

For expression of an N-terminally truncated construct of Svi3-3, the *svi3-3* gene was PCR amplified using Pfu DNA polymerase with primers Svi3-3_d19f and Svi3-3_r1 and cloned into the pEXP5-NT/ TOPO vector (Invitrogen) according to the manufacturer's protocol. Transformants were selected on LA plates supplemented with 100 µg/ml ampicillin. The correctness of plasmid pEXP5-Svi3-3_d19 was confirmed by sequencing (Eurofins). The resulting plasmid encoded amino acid 20–162 of the original His-tagged Svi3-3 polypeptide (corresponding to residues 5–147 from the phage-encoded sequence) fused to an N-terminal hexahistidine tag followed by a TEV cleavage site. The sequence is numbered starting at the N terminus of the TEV-cleaved protein sequence, corresponding to an off-set of −16 in relation to previous work (*Jerlström Hultqvist et al., 2018*).

### Site-directed mutagenesis

Svi3-3-d19 mutants Y58F, E69Q, E69A, and E105Q were generated by site-directed mutagenesis of pEXP5-Svi3-3_d19 using the QuickChange II protocol (Stratagene) using the primers listed in *Supplementary file 1*-table 3. Mutations in the plasmids pEXP5-Svi3-3_d19_Y58F, pEXP5-Svi3-3_d19_E69Q, pEXP5-Svi3-3_d19_E69A, and pEXP5-Svi3-3_d19_E105Q were confirmed by DNA sequencing.

### Protein expression and purification

Expression plasmids were transformed into BL21-AI cells and plated on LA plates containing 50 µg/ ml ampicillin and 0.1% glucose. For protein expression, 5 ml overnight culture in LB containing 50 µg/ml ampicillin and 0.1% glucose was used to inoculate 800 ml LB medium with the same composition and incubated at 37°C with shaking. When $OD_{600}$ reached 0.9, expression was induced with 0.2% L-arabinose and the culture was further incubated at 37°C for 4 hr before harvest by centrifugation. The cell pellet was resuspended in buffer A (50 mM Tris-HCl pH 7.5, 300 mM NaCl, 20 mM imidazole, 5 mM BME) including cOmplete EDTA-free protease inhibitor (Roche) and subjected to lysis by sonication. After centrifugation at 30,000 × g for 30 min, the supernatant lysate was clarified by filtration through a 0.45 µm syringe filter, loaded to a gravity column containing pre-equilibrated Ni Sepharose 6 Fast Flow (GE Healthcare) and incubated under slow rotation for 10 min at 4°C. The column was washed extensively with buffer A supplemented with 20 mM imidazole, and the His-

tagged protein was eluted with buffer A containing 500 mM imidazole. Protein-containing fractions were loaded onto a HiLoad 16/60 Superdex 75 pg column equilibrated with buffer B (25 mM Tris-HCl, 150 mM NaCl, pH 8.0). WT Svi3-3_d19, E69Q, E69A, and E105Q mutants eluted as trimers, while the Y58F mutant eluted mainly as monomer. Peak fractions were pooled and concentrated to 2 mg/ml. To cleave off the His-tag, the protein was incubated at 4°C overnight with a 1:10 molar ratio of TEV$_{SH}$ protease (*van den Berg et al., 2006*). The cleavage reaction was passed through Ni-Sepharose before being loaded onto a HiLoad 16/60 Superdex 75 pg column equilibrated in buffer B. Peak fractions were concentrated to 10 mg/ml for further use.

T3 SAMase was produced by in vitro transcription–translation as previously described (*Jerlström Hultqvist et al., 2018*).

## Crystallization, data collection, and structure determination

Crystallization was done using the sitting-drop vapor diffusion method at room temperature (293 K). Crystals grew in 2–10 days in drops containing 1 µl Svi3-3_d19 (10 mg/ml, with or without 5 mM SAH/SAM) and 1 µl of reservoir solution containing 0.4–0.6 M ammonium phosphate. Crystals were cryo-protected in reservoir solution supplemented with 1.5 M proline and vitrified in liquid nitrogen for data collection. All data were collected at ESRF beamline ID23-1 at 100 K and processed with XDS (*Kabsch, 2010*). The Svi3-3_d19 structure with SAM was solved with ab initio methods using Arcimboldo_lite (*Rodríguez et al., 2009*) run on the National Supercomputer Center (NSC) in Linköping and a 15 amino acid helix as search model (*McCoy et al., 2007*). The structure was manually rebuilt in Coot (*Emsley et al., 2010*) and refined using phenix.refine (*Afonine et al., 2012*). Statistics for data collection and refinement are summarized in *Table 1*. All structure figures were prepared using PyMol (*Schrödinger LLC, 2020*).

## SAXS

SEC-SAXS data for Svi3-3_d19 samples were collected at the Diamond Light Source on beamline B21. In-line SEC-SAXS was performed using an Agilent 1200 HPLC system connected Shodex KW403 column. Data were recorded on a Pilatus 2M detector with a fixed camera length of 4.014 m and 12.4 keV energy allowing the collection of the angular range q between 0.0038 and 0.42 Å$^{-1}$.

His$_6$-tagged Svi3-3_d19 samples at 10–13 mg/ml concentration with and without 5 mM SAM were loaded onto the size exclusion chromatography (SEC) column previously equilibrated in 25 mM Tris-HCl pH 8.0, 150 mM NaCl. Initial buffer subtraction and data processing was performed using ScÅtter (*Förster et al., 2010*). Further data analysis was performed with Primus (*Konarev et al., 2003*) and SAXSMoW (*Piiadov et al., 2019*).

## Activity assay

The activity of WT and mutant versions of the N-terminally truncated Svi3-3 construct was determined according to the previously published discontinuous assay (*Jerlström Hultqvist et al., 2018*), by separation of SAM and MTA using cation exchange chromatography. All experiments were conducted at 25°C and the enzyme concentration and duration of the experiment were adjusted to the level of activity, ranging from 50 to 500 nM enzyme and 10 min to 1 week incubation. The experiments were done in biological duplicates (two separately purified batches of each protein) and technical duplicates (two independently pipetted and measured enzymatic reactions).

## Differential scanning fluorimetry

The protocol was adopted from *Niesen et al., 2007*. Each 25 µl reaction consisted of 20 µM WT or mutant Svi3-3 in 25 mM HEPES (4–2-hydroxyethyl- 1-piperazineethanesulfonic acid), 150 mM NaCl, 0.2 µl 50× SYPRO orange dye and 0–2500 µM of SAM or SAH or 0–600 µM dcSAM mix. Reactions were done in technical triplicates (for dcSAM technical duplicates) in a BioRad CFx connect real-time system and subjected to a temperature gradient from 15°C to 95°C with an increment of 0.2°C per 30 s.

## Structure-guided sequence alignment

Structure-guided sequence alignment was performed with PROMALS3D (*Pei et al., 2008*) and manually edited. The sequence alignment figure was prepared using ESPript (*Gouet et al., 2003*).

## Inserting different gene variants encoding T3 SAMases on the chromosome

The different gene variants encoding the WT and mutant T3 SAMase were inserted on the chromosome of an $\Delta ilvA$ auxotrophic mutant of *E. coli* K-12 MG1655 by λ-red recombineering as previously described (*Datsenko and Wanner, 2000*; *Koskiniemi et al., 2011*). Each of the variants was used to replace the *araBAD* operon, so that the expression for these was under the control of the p*BAD*, the native promoter for the *araBAD* operon. Briefly, the first step involved replacing the *araBAD* operon in an $\Delta ilvA$ *E. coli* K-12 MG1655 strain by *cat-sacB-yfp* cassette. The cassette was PCR amplified using the primers araBAD_cat_sacB_F and araBAD_cat_sacB_R (*Supplementary file 1*-table 3). Native T3 SAMase and variants with mutations at E67Q and E68Q were PCR amplified using specific primers that contained homologies surrounding the *araBAD* operon at the 5' end followed by sequences that allowed amplification of the T3 SAMase variants from the respective plasmids (ara_-t3samF, ara_t3samR). PCR products were purified, DNA was transformed into the strain containing the *cat-sacB-yfp* cassette at the *araBAD* location, and transformants were selected on sucrose plates. Variants were confirmed by Sanger sequencing (using test_primer_F, test_primer_R, *Supplementary file 1*-table 3).

## In vivo complementation of $\Delta ilvA$ mutant with different variants of Svi3-3, Orf1, and T3 SAM hydrolase variants

Two different approaches were used to determine if the different variants of Svi3-3, Orf1, and T3 SAMase could complement the $\Delta ilvA$ auxotrophic mutant. In cases where the variant was present on a plasmid, the plasmid was transformed into the $\Delta ilvA$ auxotrophic mutant and was selected on LA-ampicillin (50 µg/ml) or LA-chloramphenicol (15 µg/ml) plates. Svi-3–3 and Orf1 were both cloned on a high copy number plasmid pCA24N –gfp, while T3 SAM hydrolase was cloned on an intermediate copy number plasmid pRD2. In each case the genes were placed under the IPTG-inducible promoter PLlacO. The transformants were then re-streaked, and the re-streaked colonies were tested for growth on M9-Glucose (0.2%) minimal media plates, with or without IPTG. On each test plate, the $\Delta ilvA$ auxotrophic mutant containing the empty vector was used as a negative control.

The same approach was used to test functionality of the T3 SAMase variants that were present on the chromosome, with induction being obtained using different concentrations of L-arabinose (0%, 0.01%, 0.05%, and 0.1%).

## In vivo rescue assay

E69A and E69Q variants were constructed for full-length Svi3-3 and Svi3-3_d19 to test for their ability to rescue the $\Delta ilvA$ mutant (*Jerlström Hultqvist et al., 2018*). Synthetic genes (Eurofins) containing the desired mutations and *Kpn*1 and *Xba*1 cleavage sites were cleaved-out of the original vector and ligated into a modified version of the pCA24N plasmid, purified on a spin-column and transformed directly into the $\Delta ilvA$ mutant. Plasmids were extracted from isolated colonies and sequenced to confirm the correct sequence. The respective clones were then checked for their ability to grow on minimal glucose plates. All in vivo complementation experiments were done in biological duplicates.

## Molecular dynamics

MD simulations of the Svi3-3 trimer with and without SAM in the active sites were performed with Desmond (*Bowers et al., 2006*; *Schrödinger, 2018*) using the OPLS3 force field (*Harder et al., 2016*; *Jorgensen et al., 1996*; *Jorgensen and Tirado-Rives, 1988*; *Shivakumar et al., 2010*). The Svi3-3 crystal structure in complex with SAH was used as starting conformation for the simulations where missing hydrogens were automatically added using the protein preparation wizard tool in Maestro (*Sastry et al., 2013*). The native substrate SAM was docked to the active site using Glide (*Friesner et al., 2006*; *Friesner et al., 2004*; *Halgren et al., 2004*; *Schrödinger, 2018*). The docking grid was generated with the OPLS3 force field centered on the SAH inhibitor bound to the active site of the Svi3-3 crystal structure. Due to an error in our library file, the force field parameters of the base were taken from tubercidin instead of adenine, but the charge distributions of these two bases are almost identical. The Maestro system builder (*Schrödinger, n.d.*) was used to solvate the Svi3-3 trimer with TIP3P (*Jorgensen et al., 1983*) water molecules in an orthorhombic box with buffer

distances of 10 Å to the boundary on all sides. The system was neutralized by addition of $Na^+$ ions and the final simulation box consisted of 60,972 atoms. A total of 100 ns MD simulation at 298 K was run in the NPT ensemble using the reference system propagator algorithm (RESPA) time stepping scheme (*Tuckerman et al., 1991*) with time steps of 2 fs for bonded terms, 2 fs for van der Waals and short-range electrostatic interactions, and 6 fs for long-range electrostatic interactions. Short-range Coulomb interactions were treated with a cutoff radius of 9 Å. Long-range interactions were treated with the smooth Particle Mesh Ewald method (*Darden et al., 1993*) with a tolerance of $10^{-9}$. The NPT ensemble was calculated with the Nose-Hoover chain thermostat method (*Hoover, 1985*; *Nosé, 1984*), using a relaxation time of 1 ps, and the Martyna-Tobias-Klein barostat method (*Martyna et al., 1994*), using isotropic coupling with a relaxation time of 2 ps.

## Computational reaction mechanism investigations

The catalytic mechanism of Svi3-3 was investigated with DFT calculations. A cluster model (*Figure 6—figure supplement 2B*) was generated from a snapshot of the equilibrated Svi3-3 X-ray structure taken from the MD simulation described above (*Figure 6—figure supplement 2A*). This snapshot is a typical representative displaying the key interactions shown in *Figure 6—figure supplement 1*. The active site model was composed of the backbone atoms of Val17, Gly18, Leu19, Asn20, and Val21 chopped at the N and C termini, Tyr58 and Glu69 chopped at the CA position, and Gln104, Glu105, and Ser106 (with backbone) chopped at the N and C-termini. A smaller substrate mimicking SAM was used for the DFT calculations. Here, adenosine was deleted from SAM, resulting in 2-ammonio-4-((R)-ethyl(methyl)sulfonio)butanoate, or S-ethylmethionine (SEM). In addition, a total of six water molecules from the MD snapshot were included in the cluster model, which includes the first two solvation shells of the reaction center. To account for the steric effect of the surrounding parts of the protein, atoms in the chopped positions were kept fixed to their original positions (*Figure 6—figure supplement 2*). The final model after addition of methyl groups to the chopped protein positions consisted of 194 atoms (716 electrons). In general, cluster models comprising ~200 atoms have been found to be sufficiently reliable for mechanistic investigations, provided they include all key protein residues involved in the reaction, and have also been shown to be relatively insensitive to dielectric effects (*Siegbahn and Himo, 2009*). However, as noted above, ion-pair configurations with charge separation over a large distance may be disfavored, and we thus also recalculated the reaction energetics with the system constrained to the standard protonation state of Tyr58 and Glu105, which was also the initial configuration for the optimization.

All DFT calculations were performed using the Gaussian 09 (*Frisch et al., 2009*) package. Geometry optimizations and frequency calculations were computed with the B3LYP functional (*Becke, 1993*) and the 6-31G(d,p) basis set. Dispersion effects were included in all calculations using Grimme's B3LYP-D3 method (*Grimme et al., 2011*; *Grimme et al., 2010*). Intrinsic reaction coordinate calculations were performed in both directions from the transition state to verify that the correct minima are connected. Solvent effects were obtained by single-point calculations on the optimized stationary points with the solvent model based on density (SMD) (*Marenich et al., 2009*). Electronic energies were calculated from single-point calculations on the optimized geometries (RS, TS, and PS) at the b3lyp/6-311G+(2d,2p) level of theory. The final reported energies are the electronic energies with the large basis set corrected for zero-point energies (ZPE) and solvent effects in kcal/mol.

## Production of SAM decarboxylase

*E. coli* K-12 MG1655 pCA24N:*speD* from the ASKA library (*Kitagawa et al., 2005*) encoding the hexahistidine tagged SAM decarboxylase (SDC) enzyme were inoculated in LB containing 34 μg/ml chloramphenicol. Five milliliters of saturated culture was used to inoculate 800 ml LB medium with the same composition and incubated at 37°C until $OD_{600}$ reached 0.6. Expression was induced with 0.5 mM IPTG and the culture was incubated overnight at 20°C. The cells were harvested by centrifugation at 4°C. The pellet was washed in 25 mM Tris pH 8, 150 mM NaCl, and pelleted again at 8°C. The cells were resuspended in buffer A (50 mM Tris-HCl pH 8, 300 mM NaCl, 10 mM $MgSO_4$, 5 mM BME) including cOmplete EDTA-free protease inhibitor (Roche) and subjected to lysis by sonication. After centrifugation at 30,000 × g for 30 min, the supernatant was clarified by filtration through a 0.45-μm-syringe filter, loaded in a gravity column containing 2 ml of pre-equilibrated Ni-sepharose (GE Healthcare), and incubated under slow rotation for 30 min at 8°C. The column was washed

extensively with buffer A supplemented with 20 mM imidazole, and the His-tagged protein was eluted with buffer A containing 500 mM imidazole. Protein-containing fractions were loaded onto a HiLoad 16/60 Superdex 200 column previously equilibrated with buffer B (25 mM Tris-HCl pH 8, 150 mM NaCl, 10 mM $MgSO_4$, 5 mM BME). Peak fractions were pooled and concentrated to 5 mg/ml.

## SAM decarboxylation

0.5 or 2 mM SAM was incubated with 20 µM SDC for 2 hr at 37°C in reaction buffer (20 mM HEPES pH 7, 50 mM KCl, 10 mM $MgSO_4$). After the incubation, 40 µl of the reaction was quenched with the same volume of quenching buffer (50 mM citrate pH 2.6). The rest of the reaction was filtered through a 3 kDa cutoff concentrator to remove the enzyme. The flow-through was collected and the concentration dcSAM + SAM + MTA was determined by the absorbance at 260 nm to 0.32 mM and 1.1 mM. The fraction of dcSAM was determined using ion exchange chromatography to 77% in the lower-concentration reaction and 72% in the higher-concentration reaction.

## dcSAM assay

0.32 mM dcSAM mix (77% dcSAM) was incubated with 0.1 µM Svi3-3 in reaction buffer at 37°C. As a control, dcSAM without Svi3-3 was incubated for the same time. SAM (0.32 mM) was incubated with 0.1 µM Svi3-3 and as a control only SAM (0.32 mM) was incubated for the same period of time. Samples of 40 µl were quenched with the same volume of quenching buffer after 0.5 and 23 hr. Samples were analyzed with the same cation exchange method as for the standard activity assay (*Jerlström Hultqvist et al., 2018*) but using a linear buffer gradient over seven column volumes.

## TLC

For TLC experiments, reactions containing 2 mM SAM and 0.20 µM Svi3-3, 0.20 µM Orf1, or 0.25 µM T3 SAMase in 50 mM NaPi pH 7.4 were incubated for 20 min at 25°C. MTA product formation of >70% was verified using ion exchange chromatography. 5 × 2 µl reaction was loaded on a TLC Silica gel 60 $F_{254}$ plate and developed using a mobile phase of 55% n-butanol, 30% $H_2O$, 15% acetic acid. The plate was treated with BME followed by ninhydrin staining (*Basak et al., 2005*). Briefly, the dried plate was sprayed with 1% BME in acetone, heated with a hair dryer, sprayed with 0.25% ninhydrin in acetone, and heated again until spots were clearly visible.

## NMR

To analyze the reaction products by NMR spectroscopy, a 700 µl reaction mixture was prepared containing 4 mM SAM and 500 nM Svi3-3 d19 in 100 mM Na phosphate buffer pH 7.4% and 90% $D_2O$. The pH of the sample was adjusted to 7.4 after addition of SAM with 5M NaOH. The reaction was incubated for 20 min at 25°C and shock-frozen in liquid $N_2$ and stored at −80°C until measurement. Turnover of >90% of substrate was verified by ion exchange chromatography.

[1]H NMR spectra were recorded at 600.18 MHz on a Bruker Avance Neo spectrometer equipped with a TCI (CRPHe TR-[1]H and [19]F/[13]C/[15]N 5mm-EZ) cryogenic probe for samples in aqueous sodium phosphate buffer (NaPi, 100 mM, pH 7.4, solvent $D_2O$) at 25°C. Typically 64–128 scans were accumulated with a relaxation delay of 0.7 s and an acquisition time of 2.75 s, using the zg30 pulse sequence. Spectra were obtained by zero filling the recorded 32 k data points to 128 k, followed by multiplication with an exponential weighting function and Fourier transformation.

The formation of homoserine lactone in the enzymatic reaction was confirmed by comparison with the [1]H NMR spectrum of an authentic sample (from Sigma) dissolved in the same buffer, and by comparison with literature data (*Helms et al., 1988*). Homoserine lactone spectra were also recorded for various concentrations (70 mM, 17.5 mM, 4.4 mM, 1.1 mM, and 0.3 mM) and for various times after sample preparation (*Figure 9—figure supplement 3*). The spectra indicated a gradual hydrolysis of the lactone to homoserine that was identified by comparison with literature [1]H NMR data (*Jamieson et al., 2009*) and an authentic homoserine sample (*Figure 9—figure supplement 2*).

## Acknowledgements

The authors want to thank Ulrika Yngve for advice regarding TLC.

This work was supported by grants from the Knut and Alice Wallenberg foundation (Evolution of new genes and proteins) to MS, JÅ, and DIA, from the Swedish Research Council grant 2017–03827 to MS, 2017–01527 to DIA, and from the Research Council of Norway through a Centre of Excellence and project grants (grant nos. 262695 and 274858) to GVI. The authors are grateful for the access to beamlines ID23-1 at the European Synchrotron Radiation Facility (ESRF), Grenoble, France and beamline B21 at the Diamond light source, Didcot, UK. The Archimboldo calculations were performed on resources provided by the Swedish National Infrastructure for Computing (SNIC) at National Supercomputer Centre (NSC) in Linköping. Computational resources for quantum mechanics were awarded by NOTUR; grant no. nn4654k. This study made use of the NMR Uppsala infrastructure, which was funded by the Department of Chemistry – BMC and the Disciplinary Domain of Medicine and Pharmacy.

## Additional information

### Funding

| Funder | Grant reference number | Author |
| --- | --- | --- |
| Knut och Alice Wallenbergs Stiftelse | Evolution of new genes and proteins | Johan Åqvist<br>Dan Andersson<br>Maria Selmer |
| Vetenskapsrådet | 2017-03827 | Maria Selmer |
| Research Council of Norway | 262695 | Geir Villy Isaksen |
| Vetenskapsrådet | 2017–01527 | Dan Andersson |
| Research Council of Norway | 274858 | Geir Villy Isaksen |

The funders had no role in study design, data collection and interpretation, or the decision to submit the work for publication.

### Author contributions

Xiaohu Guo, Ulrich Eckhard, Investigation, Writing - review and editing, X-ray crystallography; Annika Söderholm, Investigation, Writing - original draft, Writing - review and editing, Biochemistry; Sandesh Kanchugal P, Investigation, Visualization, Writing - review and editing, X-ray crystallography, DSF; Geir V Isaksen, Investigation, Visualization, Writing - original draft, Writing - review and editing, Computational work; Omar Warsi, Investigation, Writing - original draft, Writing - review and editing, In vivo experiments; Silvia Trigüis, Investigation, Visualization, Writing - original draft, Writing - review and editing, Biochemistry; Adolf Gogoll, Investigation, Visualization, Writing - original draft, Writing - review and editing, NMR experiments; Jon Jerlström-Hultqvist, Conceptualization, Writing - review and editing; Johan Åqvist, Resources, Supervision, Funding acquisition, Writing - review and editing; Dan I Andersson, Conceptualization, Resources, Supervision, Funding acquisition, Writing - review and editing; Maria Selmer, Conceptualization, Resources, Supervision, Funding acquisition, Validation, Investigation, Visualization, Writing - original draft, Writing - review and editing

### Author ORCIDs

Annika Söderholm https://orcid.org/0000-0003-3444-5203
Sandesh Kanchugal P http://orcid.org/0000-0001-5127-9695
Geir V Isaksen https://orcid.org/0000-0001-7828-7652
Ulrich Eckhard https://orcid.org/0000-0001-5863-4514
Adolf Gogoll http://orcid.org/0000-0002-9092-261X
Dan I Andersson https://orcid.org/0000-0001-6640-2174
Maria Selmer https://orcid.org/0000-0001-9079-2774

### Decision letter and Author response

Decision letter https://doi.org/10.7554/eLife.61818.sa1
Author response https://doi.org/10.7554/eLife.61818.sa2

## Additional files

### Supplementary files

- Supplementary file 1. Supplementary tables 1–3.
- Transparent reporting form

### Data availability

Diffraction data have been deposited to the wwPDB with accession codes 6ZM9, 6ZMG and 6ZNB. SAXS data has been deposited in SASBDB with accession codes SASDJ65 and SASDJ55.

The following datasets were generated:

| Author(s) | Year | Dataset title | Dataset URL | Database and Identifier |
|---|---|---|---|---|
| Guo X, Kanchugal P S, Selmer M | 2020 | Phage SAM lyase in complex with S-methyl-5'-thioadenosine | https://doi.org/10.2210/pdb6ZM9/pdb | Worldwide Protein Data Bank, 10.2210/pdb6ZM9/pdb |
| Guo X, Kanchugal P S, Selmer M | 2020 | Phage SAM lyase in complex with S-adenosyl-L-homocysteine | https://doi.org/10.2210/pdb6ZMG/pdb | Worldwide Protein Data Bank, 10.2210/pdb6ZMG/pdb |
| Eckhard U, Kanchugal P S, Selmer M | 2020 | Phage SAM lyase in apo state | https://doi.org/10.2210/pdb6ZNB/pdb | Worldwide Protein Data Bank, 10.2210/pdb6ZNB/pdb |
| Selmer M | 2020 | S-adenosylmethionine (SAM) lyase Svi3-3 | https://www.sasbdb.org/data/SASDJ55/huonm0h0hd/ | SASDB, SASDJ55 |
| Selmer M | 2020 | S-adenosylmethionine (SAM) lyase Svi3-3 in presence of SAM | https://www.sasbdb.org/data/SASDJ65/yicszm-l64o/ | SASDB, SASDJ65 |

The following previously published datasets were used:

| Author(s) | Year | Dataset title | Dataset URL | Database and Identifier |
|---|---|---|---|---|
| Jerlström Hultqvist J, Warsi O, Knopp M, Söderholm A, Vorontsov E, Selmer M, Andersson DI | 2017 | Synthetic construct clone Svi3-7His S-adenosylmethionine hydrolase gene, complete cds | https://www.ncbi.nlm.nih.gov/nuccore/KY556687 | NCBI GenBank, KY556687 |
| Jerlström Hultqvist J, Warsi O, Knopp M, Söderholm A, Vorontsov E, Selmer M, Andersson DI | 2017 | Synthetic construct clone Svi3-3opt S-adenosylmethionine hydrolase gene, complete cds | https://www.ncbi.nlm.nih.gov/nuccore/KY556690 | NCBI GenBank, KY556690 |
| Jerlström Hultqvist J, Warsi O, Knopp M, Söderholm A, Vorontsov E, Selmer M, Andersson DI | 2017 | Synthetic construct clone Fi5-7opt S-adenosylmethionine hydrolase gene, complete cds | https://www.ncbi.nlm.nih.gov/nuccore/KY556689 | NCBI GenBank, KY556689 |

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

# Appendix 1

**Appendix 1—key resources table**

| Reagent type (species) or resource | Designation | Source or reference | Identifiers | Additional information |
|---|---|---|---|---|
| Strain, strain background (*Escherichia coli*) | BL21-AI | Invitrogen | | Recombinant protein expression |
| Strain, strain background (*Escherichia coli*) | AG1/pCA24N (gfp-)::speD | ASKA library doi:10.1093/dnares/dsi012 | JW0116-AP | |
| Strain, strain background (*Escherichia coli* K-12 MG1655) | KEIO:1693 or DA41453 | KEIO collection doi:10.1038/msb4100050 | | |
| Strain, strain background (constructed strain *E. coli* K-12 MG1655) | DA67469 | This paper | | The native T3 SAM hydrolase was inserted on the chromosome replacing the araBAD coding region and is under the control of P*araBAD* |
| Strain, strain background (constructed strain *E. coli* K-12 MG1655) | DA67467 | This paper | | The T3 SAM hydrolase variant E67Q was inserted on the chromosome replacing the araBAD coding region and is under the control of P*araBAD* |
| Strain, strain background (constructed strain *E. coli* K-12 MG1655) | DA67468 | This paper | | The T3 SAM hydrolase variant E68Q was inserted on the chromosome replacing the araBAD coding region and is under the control of P*araBAD* |
| Gene (environmental phage DNA cloned on plasmid pCA24N) | DA48932 | doi:10.1038/s41559-018-0568-5 | KY556690.1 | Resynthesized Svi3-3 gene consisting of *E. coli* optimized codons (Eurofins) was cloned into pCA24N |
| Gene (environmental phage DNA cloned on plasmid pCA24N) | DA57022 | This paper | | A variant of the resynthesized Svi3-3 (E69Q) was cloned into pCA24N |
| Gene (environmental phage DNA cloned on plasmid pCA24N) | DA57021 | This paper | | A variant of the resynthesized Svi3-3 (E69A) was cloned into pCA24N |

*Continued on next page*

*Appendix 1—key resources table continued*

| Reagent type (species) or resource | Designation | Source or reference | Identifiers | Additional information |
|---|---|---|---|---|
| Gene (environmental phage DNA cloned on plasmid pCA24N) | DA51653 | This paper | | Similar to DA50765 from *Jerlström Hultqvist et al., 2018* |
| Gene (environmental phage DNA cloned on plasmid pCA24N) | DA67997 | This paper | | A variant of the Orf1 (E50Q) was cloned into pCA24N |
| Gene (environmental phage DNA cloned on plasmid pCA24N) | DA67998 | This paper | | A variant of the Orf1 (D51N) was cloned into pCA24N |
| Gene (environmental phage DNA cloned on plasmid pRD2) | DA57899 | doi:10.1038/s41559-018-0568-5 | | T3 SAM hydrolase was cloned into pCA24N |
| Gene (environmental phage DNA cloned on plasmid pRD2) | DA58126 | This paper | | A variant of the T3 SAM hydrolase (E67Q) was cloned into pCA24N |
| Gene (environmental phage DNA cloned on plasmid pRD2) | DA58127 | This paper | | A variant of the T3 SAM hydrolase (E68Q) was cloned into pCA24N |
| Recombinant DNA reagent | pCA24N | other | AB052891 | Plasmid used as backbone in ASKA library |
| Recombinant DNA reagent | pRD2 | other | MH298521.1 | |
| Recombinant DNA reagent | pSIM5-Tet plasmid | doi:10.1111/j.1365-2958.2011.07657.x | | λ-red recombineering plasmid |
| Recombinant DNA reagent | pEXP5-NT/TOPO | Invitrogen | V96005 | |
| Recombinant DNA reagent | pEXP5-NT-Svi3-3_d19 | This paper | | Truncated variant of Svi3-3 cloned in pEXP5-NT |
| Recombinant DNA reagent | pEXP5-NT-Svi3-3_d19_Y58F | This paper | | Mutated variant of the truncated Svi3-3 cloned in pEXP5-NT |
| Recombinant DNA reagent | pEXP5-NT-Svi3-3_d19_E69Q | This paper | | Mutated variant of the truncated Svi3-3 cloned in pEXP5-NT |
| Recombinant DNA reagent | pEXP5-NT-Svi3-3_d19_E69A | This paper | | Mutated variant of the truncated Svi3-3 cloned in pEXP5-NT |
| Recombinant DNA reagent | pEXP5-NT-Svi3-3_d19_E105Q | This paper | | Mutated variant of the truncated Svi3-3 cloned in pEXP5-NT |
| Recombinant DNA reagent | pEXP5-NT-Orf1 | doi: 10.1038/s41559-018-0568-5 | | |
| Recombinant protein | *Pfu* DNA polymerase | Thermo scientific | #EP0501 | |
| Recombinant protein | TEV$_{SH}$ protease | doi: 10.1016/j.jbiotec.2005.08.006 | RRID:Addgene_125194 | |

*Continued on next page*

*Appendix 1—key resources table continued*

| Reagent type (species) or resource | Designation | Source or reference | Identifiers | Additional information |
|---|---|---|---|---|
| Chemical compound | S-(5'-Adenosyl)-L-methionine p-toluenesulfonate salt | Sigma-Aldrich | A2408 | |
| Chemical compound | S-(5'-Adenosyl)-L-homocysteine | Sigma-Aldrich | A9384 | |
| Chemical compound | 5'-Deoxy-5'-(methylthio) adenosine | Sigma-Aldrich | D5011 | |
| Chemical compound | L-Homoserine lactone hydrochloride | Sigma-Aldrich | H7890 | |
| Chemical compound | cOmplete EDTA-free protease inhibitor | Roche | Sigma-Aldrich 11873580001 | |
| Chemical compound | SYPRO Orange | Invitrogen | S6651 | |
| Chemical compound | Ninhydrin | Sigma-Aldrich | 151173 | |
| Software, algorithm | XDS | doi:10.1107/S0907444909047337 | RRID: SCR_015652 | |
| Software, algorithm | Arcimboldo_lite | doi:10.1038/nmeth.1365 | | |
| Software, algorithm | Phaser | doi:10.1107/S0021889807021206 | RRID:SCR_014219 | |
| Software, algorithm | Coot | doi:10.1107/S0907444910007493 | RRID:SCR_014222 | |
| Software, algorithm | phenix.refine | doi:10.1107/S0907444912001308 | RRID:SCR_016736 | |
| Software, algorithm | PyMol | Schrödinger LLC | Version 2.2; RRID:SCR_000305 | |
| Software, algorithm | ScÅtter | doi:10.1107/S0021889810008289 | RRID:SCR_017271 | |
| Software, algorithm | Primus | doi:10.1107/S0021889803012779 | RRID:SCR_015648 | |
| Software, algorithm | SAXSMoW | doi:10.1002/pro.3528 | RRID:SCR_018137 | |
| Software, algorithm | PROMALS3D | doi:10.1093/nar/gkn072 | RRID:SCR_018161 | |
| Software, algorithm | ESPript | doi:10.1093/nar/gkg556 | RRID:SCR_006587 | |
| Software, algorithm | Desmond | D E Shaw Research ISBN: 978-0-7695-2700-0 | RRID:SCR_014575 | |
| Software, algorithm | Maestro | Schrodinger, LLC, New York | | |
| Software, algorithm | Protein Preparation wizard | https://doi.org/10.1007/s10822-013-9644-8 | RRID:SCR_016749 | |
| Software, algorithm | Glide | https://doi.org/10.1021/jm051256o | RRID:SCR_000187 | |
| Software, algorithm | Gaussian09 | Gaussian 09, Revision E.01 | RRID:SCR_014897 | |
| Software, algorithm | OPLS3 | https://doi.org/10.1021/acs.jctc.5b00864 | | |

*Continued on next page*

*Appendix 1—key resources table continued*

| Reagent type (species) or resource | Designation | Source or reference | Identifiers | Additional information |
|---|---|---|---|---|
| Software, algorithm | Topspin | Bruker Corp. | Version 4.0.6 RRID:SCR_014227 | |
| Software, algorithm | MestReNova | Mestrelab Research S. L. | Version 14.1.2–25024 | |
| Other | Ni Sepharose 6 Fast Flow | GE Healthcare | GE17-5318-02 | |
| Other | HiLoad 16/60 Superdex 75 pg | GE Healthcare | GE28-9893-33 | |
| Other | TLC Silica gel 60 $F_{254}$ | Merck / Supelco | 1.05554.0001 | |

