## [Decision Letter]

**Acceptance summary:**

The study reported the first phage S-adenosyl methionine (SAM) degrading enzyme (SAMase) structure, in complex with a substrate analogue and the product 5'-methyl-thioadenosine (MTA). Moreover, by combining molecular dynamics simulations, quantum-mechanical calculations, thin-layer chromatography and NMR spectroscopy, the authors convincingly demonstrated that the family of enzymes are lyases, rather than hydrolases, as once believed.

**Decision letter after peer review:**

Thank you for submitting your article "Structure and mechanism of a phage-encoded SAM lyase revises catalytic function of enzyme family" for consideration by *eLife*. Your article has been reviewed by three peer reviewers, one of whom is a member of our Board of Reviewing Editors, and the evaluation has been overseen by Cynthia Wolberger as the Senior Editor. The following individuals involved in review of your submission have agreed to reveal their identity: Vincent Moliner (Reviewer #2); Inaki Tunon (Reviewer #3).

The reviewers have discussed the reviews with one another and the Reviewing Editor has drafted this decision to help you prepare a revised submission.

In general, all reviewers found that the study is comprehensive and convincing. The combination of structural, biochemical and computational studies consistently supported that the enzyme is, in fact, a lyase rather than a hydrolase. The finding is significant and should be published in *eLife*.

However, there are some concerns about the specific cluster model used in the computational analysis. In particular, there is the question of whether the unusual protonation pattern observed in the DFT calculations reflects the small size of the cluster, and to what degree does this specific protonation pattern contribute to catalysis. If the contribution is significant, then it is important to conduct additional calculations for the relevant mutant (E105Q) to explain why its catalytic activity was found to be only slightly reduced as compared to the WT enzyme.

The reviewers also suggest to clarify several computational details and the experiment with the decarboxylated SAM (dcSAM).

Finally, several typos in the manuscript have been noted.

Reviewer #1:

In this study, the authors combined a range of experimental and computational approaches to analyze the structure and function of a phage-encoded SAM lyase. The prediction from DFT study was confirmed by experimentally analyzing the product of the enzyme and use of a SAM analog without the key carboxyl group. All the evidence combined supports the notion that the enzyme is actually a lyase, rather than a hydrolase as historically annotated.

Overall, I find the study to be quite comprehensive and convincing; it is a nice demonstration of integrated computational and experimental mechanistic investigation. The only question I have is that the DFT cluster analysis seems to indicate a deprotonated Tyr58 and protonated Glu105. To what degree this is due to the neglect of the protein environment? How important is it to catalysis? If this "reverse protonation" is in fact important to catalysis, how do we explain the minor effect of the E105Q mutation found experimentally? It would be informative to analyze the effect of mutation computationally with both DFT and MD simulations.

Reviewer #2:

The manuscript reports the first phage S-adenosyl methionine (SAM) degrading enzyme (SAMase) structure, in complex with a substrate analogue and the product 5'-methyl-thioadenosine (MTA). Authors use molecular dynamics simulations, quantum-mechanical calculations, thin-layer chromatography and NMR spectroscopy to study the mechanism of action and to propose that this family of enzymes are not hydrolases (as believed from 1960s!) but lyases. In addition, sequence analysis, in vitro and in vivo mutagenesis support that T3 SAMase belongs to the same structural family and utilizes the same reaction mechanism.

In all, the subject of the study is of interest, the calculations are well performed and complement the experiments (which I cannot review) that support the conclusions. Moreover, the manuscript is well structured and written. This reviewer has just a few concerns that authors should address before the manuscript was recommended for publication. These are as follows:

1) A total of 100 ns of MD simulation was run for both apo and holo Svi3-3 to equilibrate the systems. I wonder if the time dependence of the RMSD can be reported in the SI. Also, I wonder if structures after the MD simulations can be compared in deep with the X-ray structure?

2) I have detected some typo errors that should be polished (i.e. "…Svi3-3 is most flexible in the regions…".

3) I would suggest to change the title of the section "Computational studies…." to "quantum-mechanical studies…" because the previous section of molecular dynamics is also a computational study.

4) QM study of the mechanism was carried out with a reduced cluster model, and the effect of the protein environment is mimicked with a continuum model. Authors should comment on the limitations of this method.

5) Regarding the previous comment, isn't the reduced model too small? According to Figure 6A, the QM atoms (including the reacting breaking and forming bonds) look to be too close to the interface with the continuum model they use to treat the protein environment effects.

6) A reference is missing in the third paragraph of the subsection “Phage-encoded SAMases are lyases”.

7) I would suggest to add more information in the setting up of the model for the MD simulations. For instance, how the hydrogen atoms were added (any previous pKa calculations of the titratable residues,.…).

Reviewer #3:

This paper analyzes the structure and function of a SAM degrading enzyme (SAMase) using a combination of experimental and computational techniques. The work unequivocally characterizes this protein as a SAM lyase. The set of structural, kinetic, mutational and theoretical experiments carried out on this protein are convincing. Results are also nicely presented, being easy to follow the reasoning of the authors. I am thus happy to support the publication of this nice study.

As is almost always the case, there are a number of issues that should be considered by the authors in order to improve the manuscript. Being myself a computational chemist, I will limit most of my comments to that part of the work.

1) I am not really a fan of cluster model calculations because one is limited to a single structure and then to a single reaction path. However, I must admit that this can be enough for the purpose of identifying a reasonable reaction mechanism. However, a critical point in this approach is the selection of the starting point. In this case the authors took an unusual protonation state for the reactants. If that state is not the most stable then the barrier could be artificially reduced, resulting in a reasonable value for an enzymatic reaction, as it is the case here. In my opinion, the authors need to explore other protonation states (with a neutral Tyr58 and an unprotonated Glu105) as possible reactants states and must show that their choice is the adequate one. One must take into account that the use of truncated models could bias the selection of the protonated state.

2) Related to my previous comment, did the authors carry out molecular dynamics simulations in the selected protonation state? Is that state stable too?

3) How did the authors select the initial geometry for DFT calculations? Can the results depend on this choice?

4) I have some doubts regarding the experiments carried out with decarboxylated SAM (dcSAM). Can the authors discern if the observed lack of lyase activity is due to the fact that dcSAM doesn't bind to the enzyme or to the absence of reactivity after binding?

---

## [Author Response]

Reviewer #1:[…] Overall, I find the study to be quite comprehensive and convincing; it is a nice demonstration of integrated computational and experimental mechanistic investigation. The only question I have is that the DFT cluster analysis seems to indicate a deprotonated Tyr58 and protonated Glu105. To what degree this is due to the neglect of the protein environment? How important is it to catalysis? If this "reverse protonation" is in fact important to catalysis, how do we explain the minor effect of the E105Q mutation found experimentally? It would be informative to analyze the effect of mutation computationally with both DFT and MD simulations.

Yes, this is a relevant comment and, although our cluster model is rather large with ~200 atoms, one cannot exclude the possibility that proton relay from Tyr58 to Glu105 may be due to the limited size. The problem with ion-pairs with charge separation over a relatively large distance is now discussed in detail in the Results. As a check of the effect of the proton relay, we now recalculated the reaction energetics with the proton constrained to remain on Tyr58. It turns out that this has very little effect on the barrier and reaction energy, so that the location of the proton is not really relevant for the overall unimolecular mechanism.

Reviewer #2:[…] This reviewer has just a few concerns that authors should address before the manuscript was recommended for publication. These are as follows:1) A total of 100 ns of MD simulation was run for both apo and holo Svi3-3 to equilibrate the systems. I wonder if the time dependence of the RMSD can be reported in the SI. Also, I wonder if structures after the MD simulations can be compared in deep with the X-ray structure?

Thanks for the suggestion, the time dependence of RMSD for the three chains in the Svi3-3 trimer is relative to the experimental structure is now presented in Figure 6A. Figure 6A and B together give an overview of the fluctuations during MD simulations. We found this the best way to present the comparison of the structures since every snapshot will be different.

2) I have detected some typo errors that should be polished (i.e. "…Svi3-3 is most flexible in the regions…".

Sorry, but we do not understand this comment and cannot identify a typo in the following sentence:

“Thus, 100 ns of MD simulation demonstrated that Svi3-3 is most flexible in the regions surrounding the active site and that these regions become significantly more rigid upon substrate binding.”

We re-wrote the sentence but are not sure if this helps:

“Thus, 100 ns of MD simulation demonstrated that apo Svi3-3 shows the highest flexibility in the regions surrounding the active site, and that these regions become significantly more rigid upon substrate binding.”

3) I would suggest to change the title of the section "Computational studies…." to "quantum-mechanical studies…" because the previous section of molecular dynamics is also a computational study.

Thanks for pointing this out, we have changed the title according to the suggestion.

4) QM study of the mechanism was carried out with a reduced cluster model, and the effect of the protein environment is mimicked with a continuum model. Authors should comment on the limitations of this method.

See answer to point 5.

5) Regarding the previous comment, isn't the reduced model too small? According to Figure 6A, the QM atoms (including the reacting breaking and forming bonds) look to be too close to the interface with the continuum model they use to treat the protein environment effects.

See response to reviewer 1. Further, we have now also reported the energetics with high dielectric constant (Results). A comment on the limitations of the cluster model has been added in the Materials and methods. The main conclusion is that the reaction barrier is not very sensitive to the exact location of the proton.

6) A reference is missing in the third paragraph of the subsection “Phage-encoded SAMases are lyases”.

Thanks for noticing, this has now been added.

7) I would suggest to add more information in the setting up of the model for the MD simulations. For instance, how the hydrogen atoms were added (any previous pKa calculations of the titratable residues,.…).

The hydrogen generation procedure is now described in the Materials and methods with reference to Sastry et al., 2013.

Reviewer #3:[…] As is almost always the case, there are a number of issues that should be considered by the authors in order to improve the manuscript. Being myself a computational chemist, I will limit most of my comments to that part of the work.1) I am not really a fan of cluster model calculations because one is limited to a single structure and then to a single reaction path. However, I must admit that this can be enough for the purpose of identifying a reasonable reaction mechanism. However, a critical point in this approach is the selection of the starting point. In this case the authors took an unusual protonation state for the reactants. If that state is not the most stable then the barrier could be artificially reduced, resulting in a reasonable value for an enzymatic reaction, as it is the case here. In my opinion, the authors need to explore other protonation states (with a neutral Tyr58 and an unprotonated Glu105) as possible reactants states and must show that their choice is the adequate one. One must take into account that the use of truncated models could bias the selection of the protonated state.

The point is well taken and we are aware of the limitations of cluster models. However, they are very useful for exploring different mechanisms since those that are too high in energy can be confidently excluded. This is now clarified in the Materials and methods. The initial structure for the DFT optimizations had the standard protonation states. We now also examine the effect of the proton relay in the Results (see response to reviewers 1 and 2).

2) Related to my previous comment, did the authors carry out molecular dynamics simulations in the selected protonation state? Is that state stable too?

We did not do any MD with the deprotonated Tyr58. The Tyr-H2O-Glu-substrate hydrogen bonding pattern is very stable and involves strong H-bonds, so moving the hydrogen from donor to acceptor will not change this pattern and is not expected to cause any significant structural effects in the active site.

3) How did the authors select the initial geometry for DFT calculations? Can the results depend on this choice?

It is now clarified in the Materials and methods that the initial structure is a representative snapshot that obeys the H-bond populations in Figure 6—figure supplement 2.

4) I have some doubts regarding the experiments carried out with decarboxylated SAM (dcSAM). Can the authors discern if the observed lack of lyase activity is due to the fact that dcSAM doesn't bind to the enzyme or to the absence of reactivity after binding?

Most probably this is caused by dcSAM not binding to the enzyme. Addition of the SAM-dcSAM mixture that was tested in activity assays (Figure 8—figure supplement 1 in the revised manuscript) does not increase the thermal stability of Svi3-3 beyond what the remaining SAM+MTA does. Since even binding of the smaller ligand MTA leads to thermal stabilization, this suggests that dcSAM does not bind to Svi3-3. We have modified the title of this paragraph to “The carboxyl group of SAM appears essential for Svi3-3 binding” and re-written the last sentence of the corresponding paragraph to make this as clear as possible:

“However, in a DSF assay, dcSAM does not induce a thermal stabilization of Svi3-3, suggesting that the carboxyl group of SAM may be required for binding to Svi3-3.”